# FATE: Fairness Attacks on Graph Learning

## Abstract

We study fairness attacks on graph learning to answer the following question: *How can we achieve poisoning attacks on a graph learning model to exacerbate the bias?* We answer this question via a bi-level optimization problem and propose a meta learning-based attacking framework named FATE. The proposed framework is broadly applicable with respect to various fairness definitions and graph learning models, as well as arbitrary choices of manipulation operations. We further instantiate FATE to attack statistical parity and individual fairness on graph neural networks. We conduct extensive experimental evaluations on real-world datasets in the task of semi-supervised node classification. The experimental results demonstrate that FATE could amplify the bias of graph neural networks with or without fairness consideration while maintaining the utility on the downstream task. We hope this paper provides insights into the adversarial robustness of fair graph learning and can shed light on designing robust and fair graph learning in future studies.

## 1 Introduction

Algorithmic fairness in graph learning has received much research attention [5, 20, 24]. Despite its substantial progress, existing studies mostly assume the benevolence of input graphs and aim to ensure that the bias would not be perpetuated or amplified in the learning process. However, malicious activities in the real world are commonplace. For example, consider a financial fraud detection system which utilizes a transaction network to classify whether a bank account is fraudulent or not [49, 45]. An adversary may manipulate the transaction network (e.g., malicious banker with access to the transaction data, theft of bank accounts to make malicious transactions), so that the graph-based fraud detection model would exhibit unfair classification results with respect to people of different demographic groups. Consequently, a biased fraud detection model may infringe civil liberty to certain financial activities and impact the well-being of an individual negatively [6]. It would also make the graph learning model fail to provide the same quality of service to people of certain demographic groups, causing the financial institutions to lose business in the communities of the corresponding demographic groups. Thus, it is critical to understand how resilient a graph learning model is with respect to adversarial attacks on fairness, which we term as *fairness attacks*.

To date, fairness attack has not been well studied. Sporadic literature often follows two strategies: (1) adversarial data point injection, which is often designed for tabular data rather than graphs [38, 33, 8, 44] or (2) adversarial edge injection, which only attacks the group fairness of a graph neural network [19]. It is thus crucial to study how to attack different fairness definitions for a variety of graph learning models.

To achieve this goal, we study the Fairness attacks on graph learning (FATE) problem. We formulate it as a bi-level optimization, where the lower-level problem optimizes a task-specific loss function to make the fairness attacks deceptive and the upper-level problem leverages the supervision signal to modify the input graph and maximize the bias function corresponding to a user-defined fairness definition. To solve the bi-level optimization problem, we propose a meta learning-based solver

(FATE), whose key idea is to compute the meta-gradient of the upper-level bias function with respect to the input graph to guide the fairness attacks. Compared with existing works, our proposed FATE framework has two major advantages. First, it is capable of attacking *any* fairness definition on *any* graph learning model, as long as the corresponding bias function and the task-specific loss function are differentiable. Second, it is equipped with the ability for either continuous or discretized poisoning attacks on the graph topology. We also briefly discuss its ability for poisoning attacks on node features in a later section.

The major contributions of this paper are summarized as follows.

- **Problem definition.** We formally define the problem of fairness attacks on graph learning (the FATE problem). Based on the definition, we formulate it as a bi-level optimization problem, whose key idea is to maximize a bias function in the upper level while minimizing a task-specific loss function for a graph learning task.

- **Attacking framework.** We propose an end-to-end attacking framework named FATE. It learns a perturbed graph topology via meta learning, such that the bias with respect to the learning results trained with the perturbed graph will be amplified.

- **Empirical evaluation.** We conduct experiments on three benchmark datasets to demonstrate the efficacy of our proposed FATE framework in amplifying the bias while being the most deceptive method (i.e., achieving the highest micro F1 score) on semi-supervised node classification.

## 2   Preliminaries and Problem Definition

**A – Notations.** Throughout the paper, we use bold upper-case letter for matrix (e.g., $\mathbf{A}$), bold lower-case letter for vector (e.g., $\mathbf{x}$) and calligraphic letter for set (e.g., $\mathcal{G}$). We use superscript $^T$ to denote the transpose of a matrix/vector (e.g., $\mathbf{x}^T$ is the transpose of $\mathbf{x}$). Regarding matrix/vector indexing, we use conventions similar to NumPy in Python. For example, $\mathbf{A}[i,j]$ is the entry of $\mathbf{A}$ at the $i$-th row and $j$-th column; $\mathbf{x}[i]$ is the $i$-th entry of $\mathbf{x}$; $\mathbf{A}[i,:]$ and $\mathbf{A}[j,:]$ are the $i$-th row and $j$-th column of $\mathbf{A}$, respectively.

**B – Algorithmic fairness.** The general principle of algorithmic fairness is to ensure the learning results would not favor one side or another.[1] Among several fairness definitions that follow this principle, group fairness [16, 18] and individual fairness [15] are the most widely studied ones. Group fairness splits the entire population into multiple demographic groups by a sensitive attribute (e.g., gender) and ensure the parity of a statistical property among learning results of those groups. For example, statistical parity, a classic group fairness definition, guarantees the statistical independence between the learning results (e.g., predicted labels of a classification algorithm) and the sensitive attribute [16]. Individual fairness suggests that similar individuals should be treated similarly. It is often formulated as a Lipschitz inequality such that distance between the learning results of two data points should be no larger than the difference between these two data points [15].

**C – Problem definition.** Existing work [19] for fairness attacks on graphs randomly injects adversarial edges so that the disparity between the learning results of two different demographic groups would be amplified. However, it suffers from three major limitations. (1) First, it only attacks statistical parity while overlooking other fairness definitions (e.g., individual fairness [15]). (2) Second, it only considers adversarial edge injection, excluding other manipulations like edge deletion or reweighting. Hence, it is essential to investigate the possibility to attack other fairness definitions on real-world graphs with an arbitrary choice of manipulation operations. (3) Third, it does not consider the utility of graph learning models while achieving the fairness attacks, resulting in performance degradation in the downstream tasks. However, an institution that applies the graph learning models are often utility-maximizing [28, 2]. Thus, a performance degradation in the utility would make the fairness attacks not deceptive from the perspective of a utility-maximizing institution.

In this paper, we seek to overcome the aforementioned limitations. To be specific, given an input graph, an optimization-based graph learning model, and a user-defined fairness definition, we aim to learn a modified graph such that a bias function of the corresponding fairness definition would be maximized for *effective* fairness attacks, while minimizing the task-specific loss function with respect to the graph learning model for *deceptive* fairness attacks. Formally, we define the problem of fairness attacks on graph learning, which is referred to as the FATE problem.

---

[1] https://www.merriam-webster.com/dictionary/fairness

**Problem 1** FATE: *Fairness Attacks on Graph Learning*

**Given:** (1) An undirected graph $\mathcal{G} = \{\mathbf{A}, \mathbf{X}\}$; (2) a task-specific loss function $l(\mathcal{G}, \mathcal{Y}, \Theta, \theta)$ where $\mathcal{Y}$ is the graph learning results, $\Theta$ is the set of learnable variables and $\theta$ is the set of hyperparameters; (3) a bias function $b(\mathbf{Y}, \Theta^*, \mathbf{F}, \theta)$ where $\Theta^* = \arg\min_{\Theta} l(\mathcal{G}, \mathcal{Y}, \Theta, \theta)$ and $\mathbf{F}$ is the matrix that contains auxiliary fairness-related information (e.g., sensitive attribute values of all nodes in $\mathcal{G}$ for group fairness, pairwise node similarity matrix for individual fairness); (4) an integer budget $B$.

**Find:** a poisoned graph $\widetilde{\mathcal{G}} = \{\widetilde{\mathbf{A}}, \widetilde{\mathbf{X}}\}$ which satisfies the following properties: (1) $d(\mathcal{G}, \widetilde{\mathcal{G}}) \leq B$ where $d(\mathcal{G}, \widetilde{\mathcal{G}})$ is the distance between the input graph $\mathcal{G}$ and the poisoned graph $\widetilde{\mathcal{G}}$ (e.g., $\|\mathbf{A}, \widetilde{\mathbf{A}}\|_{1,1}$); (2) the bias function $b(\mathbf{Y}, \Theta^*, \mathbf{F})$ is maximized for effectiveness; (3) the task-specific loss function $l\left(\widetilde{\mathcal{G}}, \mathcal{Y}, \Theta, \theta\right)$ is minimized for deceptiveness.

## 3 Methodology

In this section, we first formulate Problem 1 as a bi-level optimization problem, followed by a generic meta learning-based solver named FATE.

### 3.1 Problem Formulation

Given an input graph $\mathcal{G} = \{\mathbf{A}, \mathbf{X}\}$ with adjacency matrix $\mathbf{A}$ and node feature matrix $\mathbf{X}$, an attacker aims to learn a poisoned graph $\widetilde{\mathcal{G}} = \{\widetilde{\mathbf{A}}, \widetilde{\mathbf{X}}\}$ such that the graph learning model will be maximally biased when trained on $\widetilde{\mathcal{G}}$. In this work, we consider the following settings for the attacker.

**The goal of the attacker.** The attacker aims to amplify the bias of the graph learning results output by a victim graph learning model. And the bias to be amplified is a choice made by the attacker based on which fairness definition the attacker aims to attack.

**The knowledge of the attacker.** Following similar settings in [19], we assume the attacker has access to the adjacency matrix, the feature matrix of the input graph, and the sensitive attribute of all nodes in the graph. For a (semi-)supervised learning problem, we assume that the ground-truth labels of the training nodes are also available to the attacker. For example, for a graph-based financial fraud detection problem, the malicious banker may have access to the demographic information (i.e., sensitive attribute) of the account holders and also know whether some bank accounts are fraudulent or not, which are the ground-truth labels for training nodes. Similar to [51, 52, 19], the attacker has no knowledge about the parameters of the victim model. Instead, the attacker will perform a gray-box attack by attacking a surrogate graph learning model.

**The capabilitiy of the attacker.** The attacker is able to perturb up to $B$ edges/features in the graph (i.e., $\|\mathbf{A} - \widetilde{\mathbf{A}}\|_{1,1} \leq B$ or $\|\mathbf{X} - \widetilde{\mathbf{X}}\|_{1,1} \leq B$).

Based on that, we formulate Problem 1 as a bi-level optimization problem as follows.

$$
\widetilde{\mathcal{G}} = \arg\max_{\mathcal{G}} \; b\left(\mathbf{Y}, \Theta^*, \mathbf{F}\right)
$$
$$
\text{s.t.} \quad \Theta^* = \arg\min_{\Theta} l\left(\mathcal{G}, \mathbf{Y}, \Theta, \theta\right), \; d\left(\mathcal{G}, \widetilde{\mathcal{G}}\right) \leq B \tag{1}
$$

where the lower-level problem learns an optimal surrogate graph learning model $\Theta^*$ by minimizing $l\left(\mathcal{G}, \mathbf{Y}, \Theta, \theta\right)$, the upper-level problem finds a poisoned graph $\widetilde{\mathcal{G}}$ that could maximize a bias function $b\left(\mathbf{Y}, \Theta^*, \mathbf{F}\right)$ for the victim graph learning model and the distance between the input graph and the poisoned graph $d\left(\mathcal{G}, \widetilde{\mathcal{G}}\right)$ is constrained to satisfy the setting about the budgeted attack. Note that Eq. (1) is applicable to attack *any* fairness definition on *any* graph learning model, as long as the bias function $b\left(\mathbf{Y}, \Theta^*, \mathbf{F}\right)$ and the loss function $l\left(\mathcal{G}, \mathbf{Y}, \Theta, \theta\right)$ are differentiable.

**A – Lower-level optimization problem.** A wide spectrum of graph learning models are essentially solving an optimization problem. Take the graph convolutional network (GCN) [26] as an example. It learns the node representation by aggregating information from its neighborhood, i.e., message passing. Mathematically, for an $L$-layer GCN, the hidden representation at $k$-th layer can be represented as $\mathbf{E}^{(k)} = \sigma\left(\widehat{\mathbf{A}}\mathbf{E}^{(k-1)}\mathbf{W}^{(k)}\right)$ where $\sigma$ is a nonlinear activation function (e.g., ReLU), $\widehat{\mathbf{A}} = \mathbf{D}^{-1/2}\left(\mathbf{A} + \mathbf{I}\right)\mathbf{D}^{-1/2}$ with $\mathbf{D}$ being the degree matrix of $(\mathbf{A} + \mathbf{I})$ and $\mathbf{W}^{(k)}$ is the learnable weight matrix of the $k$-th layer. Then the lower-level optimization problem aims to learn the set

136    of parameters $\Theta^* = \{\mathbf{W}^{(k)} | k = 1, \ldots, L\}$ that could minimize a task-specific loss function (e.g.,
137    cross-entropy loss for semi-supervised node classification). For more examples of graph learning
138    models from the optimization perspective, please refers to Appendix A.

139    **B – Upper-level optimization problem.** To attack the fairness aspect of a graph learning model, we
140    aim to maximize a differentiable bias function $b\left(\mathbf{Y}, \Theta^*, \mathbf{F}\right)$ with respect to a user-defined fairness
141    definition in the upper-level optimization problem. For example, for statistical parity [16], the fairness-
142    related auxiliary information matrix $\mathbf{F}$ can be defined as the one-hot demographic membership matrix,
143    where $\mathbf{F}[i, j] = 1$ if and only if node $i$ belongs to $j$-th demographic group. Then the statistical parity
144    is equivalent to the statistical independence between the learning results $\mathbf{Y}$ and $\mathbf{F}$. Based on that,
145    existing studies propose several differentiable measurements of the statistical dependence between $\mathbf{Y}$
146    and $\mathbf{F}$ as the bias function. For example, Bose et al. [5] use mutual information $I(\mathbf{Y}; \mathbf{F})$ as the bias
147    function; Prost et al. [35] define the bias function as the Maximum Mean Discrepancy $MMD\left(\mathcal{Y}_0, \mathcal{Y}_1\right)$
148    between the learning results of two different demographic groups $\mathcal{Y}_0$ and $\mathcal{Y}_1$.

149    ## 3.2 The FATE Framework

150    To solve Eq. (1), we propose a generic attacking framework named FATE to learn the poisoned graph.
151    The key idea is to view Eq. (1) as a meta learning problem, which aims to find suitable hyperparameter
152    settings for a learning task [3], and treat the graph $\mathcal{G}$ as a hyperparameter. With that, we learn the
153    poisoned graph $\widetilde{\mathcal{G}}$ using the meta-gradient of the bias function $b\left(\mathbf{Y}, \Theta^*, \mathbf{F}\right)$ with respect to $\mathcal{G}$. In the
154    following, we introduce two key parts of FATE in details, including meta-gradient computation and
155    graph poisoning with meta-gradient.

156    **A – Meta-gradient computation.** The key term to learn the poisoned graph is the meta-gradient of
157    the bias function with respect to the graph $\mathcal{G}$. Before computing the meta-gradient, we assume that
158    the lower-level optimization problem converges in $T$ epochs. Thus, we first pre-train the lower-level
159    optimization problem by $T$ epochs to obtain the optimal model $\Theta^* = \Theta^{(T)}$ before computing the
160    meta-gradient. The training of the lower-level optimization problem can also be viewed as a dynamic
161    system with the following updating rule

$$\Theta^{(t+1)} = \mathrm{opt}^{(t+1)}\left(\mathcal{G}, \Theta^{(t)}, \theta, \mathbf{Y}\right), \ \forall t \in \{1, \ldots, T\} \tag{2}$$

162    where $\Theta^{(1)}$ refers to $\Theta$ at initialization, $\mathrm{opt}^{(t+1)}(\cdot)$ is an optimizer that minimizes the lower-level
163    loss function $l\left(\mathcal{G}, \mathbf{Y}, \Theta^{(t)}, \theta\right)$ at $(t + 1)$-th epoch. From the perspective of the dynamic system,
164    by applying the chain rule and unrolling the training of lower-level problem with Eq. (2), the
165    meta-gradient $\nabla_{\mathcal{G}} b$ can be written as

$$\nabla_{\mathcal{G}} b = \nabla_{\mathcal{G}} b\left(\mathbf{Y}, \Theta^{(T)}, \mathbf{F}\right) + \sum_{t=0}^{T-2} A_t B_{t+1} \ldots B_{T-1} \nabla_{\theta^{(T)}} b\left(\mathbf{Y}, \Theta^{(T)}, \mathbf{F}\right) \tag{3}$$

166    where $A_t = \nabla_{\mathcal{G}} \Theta^{(t+1)}$ and $B_t = \nabla_{\Theta^{(t)}} \Theta^{(t+1)}$. However, Eq. (3) is computationally expensive in
167    both time and space. To further speed up the computation, we adopt a first-order approximation of
168    the meta-gradient [17] and simplify the meta-gradient as

$$\nabla_{\mathcal{G}} b \approx \nabla_{\Theta^{(T)}} b\left(\mathbf{Y}, \Theta^{(T)}, \mathbf{F}\right) \cdot \nabla_{\mathcal{G}} \Theta^{(T)} \tag{4}$$

169    Since the input graph is undirected, the derivative of the symmetric adjacency matrix $\mathbf{A}$ can be
170    computed as follows by applying the chain rule of a symmetric matrix [21].

$$\nabla_{\mathbf{A}} b \leftarrow \nabla_{\mathbf{A}} b + \left(\nabla_{\mathbf{A}} b\right)^T - \mathrm{diag}\left(\nabla_{\mathbf{A}} b\right) \tag{5}$$

171    For the node feature matrix $\mathbf{X}$, its derivative is equal to the partial derivative $\nabla_{\mathbf{X}} b$ since it is often an
172    asymmetric matrix.

173    **B – Graph poisoning with meta-gradient.** After computing the meta-gradient of the bias function
174    $\nabla_{\mathcal{G}} b$, we aim to poison the input graph guided by $\nabla_{\mathcal{G}} b$. We introduce two poisoning strategies: (1)
175    continuous poisoning and (2) discretized poisoning.

176    *Continuous poisoning attack.* The continuous poisoning attack is straightforward by reweighting
177    edges in the graph. We first compute the meta-gradient of the bias function $\nabla_{\mathbf{A}} b$, then use it to poison
178    the input graph in a gradient descent-based updating rule as follows.

$$\mathbf{A} \leftarrow \mathbf{A} - \eta \nabla_{\mathbf{A}} b \tag{6}$$

where $\eta$ is a learning rate to control the magnitude of the poisoning attack. The learning rate should satisfy $\eta \le \frac{B}{\|\nabla_\mathbf{A}\|_{1,1}}$ to ensure that constraint on the budgeted attack.

*Discretized poisoning attack.* The discretized poisoning attack aims to select a set of edges to be added/deleted. It is guided by a poisoning preference matrix defined as follows.

$$\nabla_\mathbf{A} = (\mathbf{1} - 2\mathbf{A}) \circ \nabla_\mathbf{A} b \tag{7}$$

where $\mathbf{1}$ is an all-one matrix with the same dimension as $\mathbf{A}$ and $\circ$ denotes the Hadamard product. A large positive $\nabla_\mathbf{A}[i,j]$ indicates strong preference in adding an edge if nodes $i$ and $j$ are not connected (i.e., positive $\nabla_\mathbf{A} b[i,j]$, positive $(\mathbf{1} - 2\mathbf{A})[i,j]$) or deleting an edge if nodes $i$ and $j$ are connected (i.e., negative $\nabla_\mathbf{A} b[i,j]$, negative $(\mathbf{1} - 2\mathbf{A})[i,j]$). Then, a greedy selection strategy is applied to find the set of edges $\mathcal{E}_{\text{attack}}$ to be added/deleted.

$$\mathcal{E}_{\text{attack}} = \text{topk}(\nabla_\mathbf{A}, \delta) \tag{8}$$

where $\text{topk}(\nabla_\mathbf{A}, \delta)$ selects $\delta$ entries with highest preference score in $\nabla_\mathbf{A}$. Note that, if we only want to add edges without any deletion, all negative entries in $\nabla_\mathbf{A} b$ should be zeroed out before computing Eq. (7). Likewise, if edges are only expected to be deleted, all positive entries should be zeroed out.

*Remarks.* Poisoning node feature matrix $\mathbf{X}$ follows the same steps as poisoning adjacency matrix $\mathbf{A}$ without applying Eq. (5).

**C – Overall framework.** FATE generally works as follows. (1) We first pre-train the surrogate graph learning model and get the corresponding learning model $\Theta^{(T)}$ as well as the learning results $\mathbf{Y}^{(T)}$. (2) Then we compute the meta gradient of the bias function using Eqs. (4) and (5). (3) Finally, we perform the discretized poisoning attack (Eqs. (7) and (8)) or continuous poisoning attack (Eq. (6)). A detailed pseudo-code of FATE is provided in Appendix B.

**D – Limitations.** Since FATE leverages the meta-gradient to poison the input graph, it requires the bias function $b\left(\mathbf{Y}, \Theta^{(T)}, \mathbf{F}\right)$ to be differentiable in order to calculate the meta-gradient $\nabla_\mathcal{G} b$. In Sections 4 and 5, we present a carefully chosen bias function for FATE. And we leave it for future work on exploring the ability of FATE in attacking other fairness definitions. Moreover, though the meta-gradient can be efficiently computed via auto-differentiation in many deep learning packages (e.g., PyTorch[2], TensorFlow[3]), it requires $O(n^2)$ space complexity to store the meta-gradient when attacking fairness via edge flipping. It is still a challenging open problem on how to efficiently compute the meta-gradient in terms of space. One possible remedy for discretized attack might be a low-rank approximation on the perturbation matrix formed by $\mathcal{E}_{\text{attack}}$. Since the difference between the benign graph and poisoned graph are often small and budgeted ($d\left(\mathcal{G}, \widetilde{\mathcal{G}}\right) \le B$), it is likely that the edge manipulations may be around a few set of nodes, which makes the perturbation matrix to be an (approximately) low-rank matrix.

# 4 Instantiation #1: Statistical Parity on Graph Neural Networks

Here, we instantiate FATE framework by attacking statistical parity on graph neural networks in a binary node classification problem with a binary sensitive attribute. We briefly discuss how to choose (1) the surrogate graph learning model used by the attacker, (2) the task-specific loss function in the lower-level optimization problem and (3) the bias function in the upper-level optimization problem.

**A – Surrogate graph learning model.** We assume that the surrogate model to be used by the attacker is a 2-layer linear GCN [47] with different hidden dimensions and model parameters at initialization.

**B – Lower-level loss function.** We consider a semi-supervised node classification task for the graph neural network to be attacked. Thus, the lower-level loss function is chosen as the cross entropy between the ground-truth label and the predicted label: $l\left(\mathcal{G}, \mathbf{Y}, \Theta, \theta\right) = \frac{1}{|\mathcal{V}_{\text{train}}|} \sum_{i \in \mathcal{V}_{\text{train}}} \sum_{j=1}^{c} y_{i,j} \ln \widehat{y}_{i,j}$, where $\mathcal{V}_{\text{train}}$ is the set of training nodes with ground-truth labels with $|\mathcal{V}_{\text{train}}|$ being its cardinality, $c$ is the number of classes, $y_{i,j}$ is a binary indicator of whether node $i$ belongs to class $j$ and $\widehat{y}_{i,j}$ is the prediction probability of node $i$ belonging to class $j$.

**C – Upper-level bias function.** We aim to attack statistical parity in the upper-level problem, which asks for $\mathrm{P}\left[\hat{y} = 1\right] = \mathrm{P}\left[\hat{y} = 1 | s = 1\right]$. Suppose $p\left(\widehat{y}\right)$ is the probability density function (PDF) of $\widehat{y}_{i,1}$

---

[2]https://pytorch.org/
[3]https://www.tensorflow.org/

for any node $i$ and $p\left(\widehat{y}|s=1\right)$ is the PDF of $\widehat{y}_{i,1}$ for any node $i$ belong to the demographic group with sensitive attribute value $s=1$. We observe that $\mathrm{P}\left[\hat{y}=1\right]$ and $\mathrm{P}\left[\hat{y}=1|s=1\right]$ are equivalent to the cumulative distribution functions (CDF) of $p\left(\widehat{y}<\frac{1}{2}\right)$ and $p\left(\widehat{y}<\frac{1}{2}|s=1\right)$, respectively. To estimate both $\mathrm{P}\left[\hat{y}=1\right]$ and $\mathrm{P}\left[\hat{y}=1|s=1\right]$ with a differentiable function, we first estimate their probability density functions ($p\left(\widehat{y}<\frac{1}{2}\right)$ and $p\left(\widehat{y}<\frac{1}{2}|s=1\right)$) with kernel density estimation (KDE, Definition 1).

**Definition 1** *(Kernel density estimation [7]) Given a set of $n$ IID samples $\{x_1,\ldots,x_n\}$ drawn from a distribution with an unknown probability density function $f$, the kernel density estimation of $f$ at point $\tau$ is defined as follows.*

$$\widetilde{f}\left(\tau\right) = \frac{1}{na}\sum_{i=1}^{n} f_k\left(\frac{\tau-x_i}{a}\right) \tag{9}$$

*where $\widetilde{f}$ is the estimated probability density function, $f_k$ is the kernel function and $a$ is a non-negative bandwidth.*

Moreover, we assume the kernel function in KDE is the Gaussian kernel $f_k\left(x\right) = \frac{1}{\sqrt{2\pi}}e^{-x^2/2}$. However, computing the CDF of a Gaussian distribution is non-trivial. Following [9], we leverage a tractable approximation of the Gaussian Q-function as follows.

$$Q(\tau) = F_k\left(\tau\right) = \int_{\tau}^{\infty} f_k(x)dx \approx e^{-\alpha\tau^2-\beta\tau-\gamma} \tag{10}$$

where $f_k(x) == \frac{1}{\sqrt{2\pi}}e^{-x^2/2}$ is a Gaussian distribution with zero mean, $\alpha=0.4920$, $\beta=0.2887$, $\gamma=1.1893$ [30]. The overall workflow of estimating $\mathrm{P}\left[\hat{y}=1\right]$ is as follows.

- For any node $i$, get its prediction probability $\widehat{y}_{i,1}$ with respect to class 1;
- Estimate the CDF $\mathrm{P}\left[\hat{y}=1\right]$ using a Gaussian KDE with bandwidth $a$ by $\mathrm{P}\left[\hat{y}=1\right] = \frac{1}{n}\sum_{i=1}^{n}\exp\left(-\alpha\left(\frac{0.5-\widehat{y}_{i,1}}{a}\right)^2 - \beta\left(\frac{0.5-\widehat{y}_{i,1}}{a}\right) - \gamma\right)$, where $\alpha=0.4920$, $\beta=0.2887$, $\gamma=1.1893$ and $\exp(x)=e^x$.

Note that $\mathrm{P}\left[\hat{y}=1|s=1\right]$ can be estimated with a similar procedure with minor modifications. The only modifications needed are: (1) get the prediction probability of nodes with $s=1$ and (2) compute the CDF using the Gaussian Q-function over nodes with $s=1$ rather than all nodes in the graph.

## 5 Instantiation #2: Individual Fairness on Graph Neural Networks

We provide another instantiation of FATE framework by attacking individual fairness on graph neural networks. Here, we consider the same surrogate graph learning model (i.e., 2-layer linear GCN) and the same lower-level loss function (i.e., cross entropy) as described in Section 4. To attack individual fairness, we define the upper-level bias function following the principles in [20]: the fairness-related auxiliary information matrix $\mathbf{F}$ is defined as the oracle symmetric pairwise node similarity matrix $\mathbf{S}$ (i.e., $\mathbf{F}=\mathbf{S}$), where $\mathbf{S}[i,j]$ measures the similarity between node $i$ and node $j$. Kang et al. [20] define that the overall individual bias to be $\mathrm{Tr}\left(\mathbf{Y}^T\mathbf{L_S}\mathbf{Y}\right)$. Assuming that $\mathbf{Y}$ is the output of an optimization-based graph learning model, $\mathbf{Y}$ can be viewed as a function with respect to the input graph $\mathcal{G}$, which makes $\mathrm{Tr}\left(\mathbf{Y}^T\mathbf{L_S}\mathbf{Y}\right)$ differentiable with respect to $\mathcal{G}$. Thus, the bias function $b(\cdot)$ can be naturally defined as the overall individual bias of the input graph $\mathcal{G}$, i.e., $b\left(\mathbf{Y},\Theta^*,\mathbf{S}\right) = \mathrm{Tr}\left(\mathbf{Y}^T\mathbf{L_S}\mathbf{Y}\right)$.

## 6 Experiments

### 6.1 Attacking Statistical Parity on Graph Neural Networks

**Settings.** We compare FATE with 4 baseline methods, i.e., Random, DICE [46], FA-GNN [19], under the same setting as in Section 4. That is, (1) the fairness definition to be attacked is statistical parity; (2) the downstream task is binary semi-supervised node classification with binary sensitive attributes. The experiments are conducted on 3 real-world datasets, i.e., Pokec-n, Pokec-z and Bail. Similar to existing works, we use the 50%/25%/25% splits for train/validation/test sets. For all baseline

Table 1: Effectiveness of attacking group fairness on GCN. FATE poisons the graph via both edge flipping (FATE-flip) and edge addition (FATE-add) while all other baselines poison the graph via edge addition. Higher is better (↑) for micro F1 score (Micro F1) and $\Delta_{\text{SP}}$. Bold font indicates the success of fairness attack (i.e., $\Delta_{\text{SP}}$ is increased after fairness attack) with the highest micro F1 score. Underlined cell indicates the failure of fairness attack (i.e., $\Delta_{\text{SP}}$ is decreased after fairness attack).

| Dataset | Ptb. | Random | | DICE | | FA-GNN | | FATE-flip | | FATE-add | |
|---|---|---|---|---|---|---|---|---|---|---|---|
| | | Micro F1 (↑) | $\Delta_{\text{SP}}$ (↑) | Micro F1 (↑) | $\Delta_{\text{SP}}$ (↑) | Micro F1 (↑) | $\Delta_{\text{SP}}$ (↑) | Micro F1 (↑) | $\Delta_{\text{SP}}$ (↑) | Micro F1 (↑) | $\Delta_{\text{SP}}$ (↑) |
| Pokec-n | 0.00 | 67.5 ± 0.3 | 7.1 ± 0.4 | 67.5 ± 0.3 | 7.1 ± 0.4 | 67.5 ± 0.3 | 7.1 ± 0.4 | 67.5 ± 0.3 | 7.1 ± 0.4 | 67.5 ± 0.3 | 7.1 ± 0.4 |
| | 0.05 | 68.0 ± 0.3 | 6.2 ± 0.8 | 67.6 ± 0.2 | 6.8 ± 0.3 | 67.8 ± 0.1 | 3.3 ± 0.4 | 67.9 ± 0.4 | 9.3 ± 1.2 | 67.9 ± 0.4 | 9.3 ± 1.2 |
| | 0.10 | 66.8 ± 0.8 | 7.3 ± 0.7 | 66.1 ± 0.5 | 6.6 ± 1.1 | 66.0 ± 0.2 | 11.5 ± 0.6 | 68.2 ± 0.6 | 9.8 ± 1.5 | 68.2 ± 0.6 | 9.8 ± 1.5 |
| | 0.15 | 66.7 ± 0.4 | 8.1 ± 0.4 | 65.6 ± 0.4 | 7.7 ± 0.8 | 66.0 ± 0.4 | 15.6 ± 3.0 | 68.0 ± 0.3 | 11.5 ± 1.0 | 68.0 ± 0.3 | 11.5 ± 1.0 |
| | 0.20 | 66.3 ± 0.7 | 8.6 ± 1.8 | 64.2 ± 0.4 | 3.4 ± 0.9 | 65.8 ± 0.1 | 18.4 ± 0.7 | 68.2 ± 0.5 | 12.0 ± 1.8 | 68.2 ± 0.5 | 12.0 ± 1.8 |
| | 0.25 | 66.2 ± 0.6 | 8.5 ± 0.8 | 63.4 ± 0.2 | 6.3 ± 0.8 | 66.6 ± 0.2 | 23.3 ± 0.5 | 68.3 ± 0.4 | 12.1 ± 2.1 | 68.3 ± 0.4 | 12.1 ± 2.1 |
| Pokec-z | 0.00 | 68.4 ± 0.4 | 6.6 ± 0.9 | 68.4 ± 0.4 | 6.6 ± 0.9 | 68.4 ± 0.4 | 6.6 ± 0.9 | 68.4 ± 0.4 | 6.6 ± 0.9 | 68.4 ± 0.4 | 6.6 ± 0.9 |
| | 0.05 | 68.8 ± 0.4 | 6.4 ± 0.6 | 67.4 ± 0.5 | 6.6 ± 0.3 | 68.1 ± 0.3 | 2.2 ± 0.4 | 68.7 ± 0.4 | 6.7 ± 1.4 | 68.7 ± 0.4 | 6.7 ± 1.4 |
| | 0.10 | 68.7 ± 0.3 | 8.0 ± 0.6 | 66.5 ± 0.2 | 6.3 ± 0.8 | 67.7 ± 0.4 | 13.5 ± 0.9 | 68.7 ± 0.6 | 7.5 ± 0.7 | 68.7 ± 0.6 | 7.5 ± 0.7 |
| | 0.15 | 67.9 ± 0.3 | 9.1 ± 0.8 | 65.9 ± 0.8 | 5.5 ± 1.3 | 66.6 ± 0.4 | 16.9 ± 2.6 | 69.0 ± 0.8 | 8.5 ± 1.1 | 69.0 ± 0.8 | 8.5 ± 1.1 |
| | 0.20 | 68.5 ± 0.4 | 9.3 ± 1.0 | 62.9 ± 0.7 | 8.7 ± 1.0 | 66.1 ± 0.2 | 25.4 ± 1.3 | 68.5 ± 0.6 | 8.8 ± 1.1 | 68.5 ± 0.6 | 8.8 ± 1.1 |
| | 0.25 | 68.3 ± 0.5 | 7.3 ± 0.5 | 63.9 ± 0.4 | 6.0 ± 1.0 | 65.5 ± 0.6 | 22.3 ± 2.8 | 68.5 ± 1.1 | 8.6 ± 2.5 | 68.5 ± 1.1 | 8.6 ± 2.5 |
| Bail | 0.00 | 93.1 ± 0.2 | 8.0 ± 0.2 | 93.1 ± 0.2 | 8.0 ± 0.2 | 93.1 ± 0.2 | 8.0 ± 0.2 | 93.1 ± 0.2 | 8.0 ± 0.2 | 93.1 ± 0.2 | 8.0 ± 0.2 |
| | 0.05 | 92.7 ± 0.2 | 8.1 ± 0.0 | 91.6 ± 0.2 | 8.5 ± 0.1 | 91.7 ± 0.1 | 10.0 ± 0.4 | 92.6 ± 0.1 | 8.6 ± 0.1 | 92.5 ± 0.1 | 8.6 ± 0.1 |
| | 0.10 | 92.2 ± 0.2 | 7.8 ± 0.2 | 90.3 ± 0.1 | 8.5 ± 0.1 | 90.5 ± 0.0 | 10.3 ± 0.4 | 92.4 ± 0.1 | 8.9 ± 0.1 | 92.4 ± 0.1 | 8.6 ± 0.1 |
| | 0.15 | 91.9 ± 0.2 | 7.8 ± 0.1 | 89.2 ± 0.1 | 7.7 ± 0.1 | 90.0 ± 0.2 | 8.4 ± 0.2 | 92.2 ± 0.2 | 9.1 ± 0.1 | 92.3 ± 0.1 | 9.1 ± 0.1 |
| | 0.20 | 91.6 ± 0.2 | 7.8 ± 0.1 | 88.3 ± 0.1 | 8.3 ± 0.1 | 89.7 ± 0.1 | 7.4 ± 0.4 | 92.2 ± 0.0 | 9.3 ± 0.1 | 92.3 ± 0.1 | 9.3 ± 0.2 |
| | 0.25 | 91.4 ± 0.1 | 8.3 ± 0.1 | 87.8 ± 0.0 | 7.8 ± 0.1 | 89.8 ± 0.2 | 5.2 ± 0.2 | 92.1 ± 0.1 | 9.1 ± 0.2 | 92.1 ± 0.1 | 9.1 ± 0.3 |

methods, the victim models are set to GCN [26]. For each dataset, we use a fixed random seed to learn the poisoned graph corresponding to each baseline method. Then we train the victim model 5 times with different random seeds. For fair comparison, we only attack the adjacency matrix in all experiments. Please refer to Appendix C for detailed experimental settings.

**Main results.** For FATE, we conduct fairness attacks via both edge flipping (FATE-flip in Table 5) and edge addition (FATE-add in Table 1). For all other baseline methods, edges are only added. The effectiveness of fairness attacks on GCN are presented in Tables 5. From both tables, we have the following key observations: (1) FATE-flip and FATE-add are the only methods that consistently succeeds in fairness attacks, while all other baseline methods might fail in some cases (indicated by the underlined $\Delta_{\text{SP}}$ in both tables) because of the decrease in $\Delta_{\text{SP}}$. (2) FATE-flip and FATE-add can not only amplify $\Delta_{\text{SP}}$ consistently, but also achieve the best micro F1 score on node classification, which makes FATE-flip and FATE-add more deceptive than all baseline methods. Notably, FATE-flip and FATE-add are able to even increase micro F1 score on all datasets, while other baseline methods attack the graph neural networks at the expense of utility (micro F1 score). (3) Though FA-GNN could make the model more biased in some cases, it cannot guarantee consistent success in fairness attacks on all three datasets as shown by the underlined $\Delta_{\text{SP}}$ in both tables. All in all, our proposed FATE framework is the framework that consistently succeeds in fairness attacks while being the most deceptive (i.e., highest micro F1 score).

**Effect of the perturbation rate.** From Table 1, we have the following observations. First, $\Delta_{\text{SP}}$ tends to increase when the perturbation rate increases, which demonstrates the effectiveness of FATE-flip and FATE-add for attacking fairness. Though in some cases $\Delta_{\text{SP}}$ might have a marginal decrease, FATE-flip and FATE-add still successfully attack the fairness compared with GCN trained on the benign graph by being larger to the $\Delta_{\text{SP}}$ when perturbation rate (Ptb.) is 0. Second, FATE-flip and FATE-add are deceptive, meaning that the micro F1 scores is close to or even higher than the micro F1 scores on the benign graph compared with the corresponding metrics trained . In summary, across different perturbation rates, FATE-flip and FATE-add are both effective, i.e., amplifying more bias with higher perturbation rate, and deceptive, i.e., achieving similar or even higher micro F1 score.

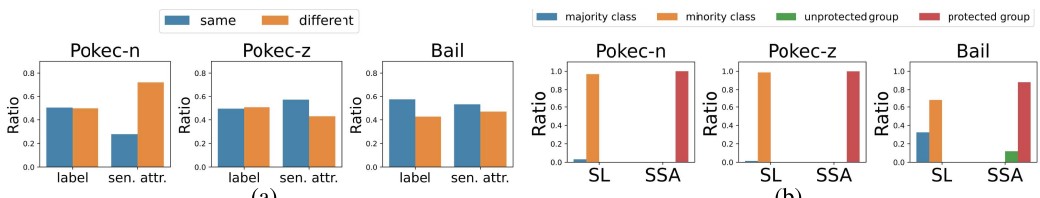

Figure 1: Attacking statistical parity with FATE-flip. (a) Ratios of flipped edges that connect two nodes with same/different label or sensitive attribute (sens. attr.). (b) SL (abbreviation for same label) refers to the ratios of flipped edges whose two endpoints are both from the same class. SSA (abbreviation for same sensitive attribute) refers to the ratios of manipulated edges whose two endpoints are both from the same demographic group. Majority/minority classes are determined by splitting the training nodes based on their class labels. The protected group is the demographic group with fewer nodes.

**Analysis on the manipulated edges.** Here, we aim to characterize the properties of edges that are flipped by FATE (i.e., FATE-flip) in attacking statistical parity. The reason to only analyze FATE-flip is that the majority of edges manipulated by FATE-flip on all three datasets is by addition (i.e., flipping from non-existing to existing). Figure 1b suggests that, if the two endpoints of an manipulated edge share the same class label or same sensitive attribute value, these two endpoints are most likely from the minority class and protected group. Combining Figures 1a and 1b, FATE would significantly increase the number of edges that are incident to nodes in the minority class and/or protected group.

**More experimental results.** Due to the space limitation, we defer more experimental results on attacking statistical parity on graph neural networks in Appendix D. More specifically, we present the performance evaluation under different metrics, i.e., Macro F1 and AUC, as well as the effectiveness of FATE with a different victim model, i.e., FairGNN [11], which ensures statistical parity.

## 6.2 Attacking Individual Fairness on Graph Neural Networks

**Settings.** To showcase the ability of FATE on attacking the individual fairness (Section 5), we further compare FATE with the same set of baseline methods (Random, DICE [46], FA-GNN [19]) on the same set of datasets (Pokec-n, Pokec-z, Bail). We follow the settings as in Section 5. We use the 50%/25%/25% splits for train/validation/test sets with GCN [26] being the victim model. For each dataset, we use a fixed random seed to learn the poisoned graph corresponding to each baseline method. Then we train the victim model 5 times with different random seeds. And each entry in the oracle pairwise node similarity matrix is computed by the cosine similarity of the corresponding rows in the adjacency matrix. That is, $\mathbf{S}[i,j] = \cos(\mathbf{A}[i,:], A[j,:])$, where $\cos()$ is the function to compute cosine similarity. For fair comparison, we only attack the adjacency matrix in all experiments. Please refer to Appendix C for detailed experimental settings.

**Main results.** Similarly, we test FATE with both edge flipping (FATE-flip in Table 2) and edge addition (FATE-add in Table 2), while all other baseline methods only add edges. From Table 2, we have two key observations. (1) FATE-flip and FATE-add are effective: they are the only methods that could consistently attack individual fairness whereas all other baseline methods mostly fail to attack individual fairness. (2) FATE-flip and FATE-add are deceptive: they achieve comparable or even better utility on all datasets compared with the utility on the benign graph. Hence, FATE framework is able to achieve effective and deceptive attacks to exacerbate individual bias.

**Effect of the perturbation rate.** From Table 2, we obtain similar observations as in Section 6.1 for Bail dataset. While for Pokec-n and Pokec-z, the correlation between the perturbation rate (Ptb.) and the individual bias is weaker. One possible reason is that: for Pokec-n and Pokec-z, the discrepancy between the oracle pairwise node similarity matrix and the benign graph is larger. Since the individual bias is computed using the oracle pairwise node similarity matrix rather than the benign/poisoned adjacency matrix, higher perturbation rate to poison the adjacency matrix may have less impact on the computation of individual bias.

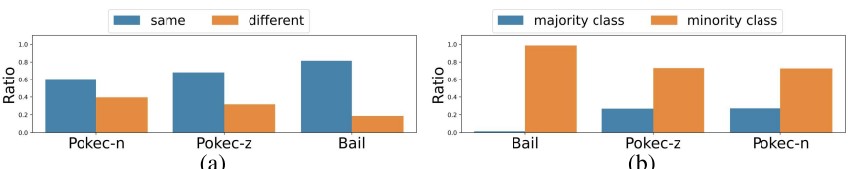

Figure 2: Attacking individual fairness with FATE-flip. (a) Ratios of flipped edges that connect two nodes with same/different label. (b) Ratios of flipped edges whose two endpoints are both from the majority/minority class. Majority/minority classes are formed by splitting the training nodes based on their class labels.

**Analysis on the manipulated edges.** Similarly, since the majority of edges manipulated by FATE-flip is through addition, we only analyze FATE-flip here. From Figure 2, we can find out that FATE will manipulate edges from the same class (especially from the minority class). In this way, FATE would find edges that could increase individual bias and improve the utility of the minority class in order to make the fairness attack deceptive.

**More experimental results.** Due to the space limitation, we defer more experimental results on attacking individual fairness on graph neural networks in Appendix E. More specifically, we present the performance evaluation under different metrics, i.e., Macro F1 and AUC, as well as

Table 2: Effectiveness of attacking individual fairness on GCN. FATE poisons the graph via both edge flipping (FATE-flip) and edge addition (FATE-add) while all other baselines poison the graph via edge addition. Higher is better (↑) for micro F1 score (Micro F1) and InFoRM bias (Bias). Bold font indicates the success of fairness attack (i.e., bias is increased after attack) with the highest micro F1 score. Underlined cell indicates the failure of fairness attack (i.e., $\Delta_{SP}$ is decreased after attack).

| Dataset | Ptb. | Random | | DICE | | FA-GNN | | FATE-flip | | FATE-add | |
|---|---|---|---|---|---|---|---|---|---|---|---|
| | | Micro F1 (↑) | Bias (↑) | Micro F1 (↑) | Bias (↑) | Micro F1 (↑) | Bias (↑) | Micro F1 (↑) | Bias (↑) | Micro F1 (↑) | Bias (↑) |
| Pokec-n | 0.00 | 67.5 ± 0.3 | 0.9 ± 0.2 | 67.5 ± 0.3 | 0.9 ± 0.2 | 67.5 ± 0.3 | 0.9 ± 0.2 | 67.5 ± 0.3 | 0.9 ± 0.2 | 67.5 ± 0.3 | 0.9 ± 0.2 |
| | 0.05 | 67.6 ± 0.3 | 1.6 ± 0.3 | 66.9 ± 0.3 | 1.6 ± 0.2 | **67.8 ± 0.5** | **1.9 ± 0.2** | 67.8 ± 0.3 | 1.2 ± 0.4 | 67.6 ± 0.3 | 1.5 ± 0.6 |
| | 0.10 | 67.2 ± 0.5 | 1.4 ± 0.3 | 65.3 ± 0.7 | 1.1 ± 0.1 | 67.4 ± 0.4 | 1.2 ± 0.2 | **67.9 ± 0.4** | **1.3 ± 0.3** | 67.7 ± 0.4 | 1.6 ± 0.4 |
| | 0.15 | 67.2 ± 0.3 | 1.2 ± 0.4 | 63.9 ± 0.6 | 1.1 ± 0.2 | 66.1 ± 0.3 | 1.5 ± 0.3 | **67.8 ± 0.4** | **1.2 ± 0.2** | 67.6 ± 0.2 | 1.1 ± 0.3 |
| | 0.20 | 66.6 ± 0.3 | 1.1 ± 0.2 | 63.8 ± 0.1 | 0.8 ± 0.1 | 65.7 ± 0.6 | 1.5 ± 0.3 | 67.3 ± 0.4 | 1.1 ± 0.3 | **68.2 ± 1.0** | **1.7 ± 0.8** |
| | 0.25 | 66.7 ± 0.3 | 1.3 ± 0.4 | 62.5 ± 0.4 | 0.6 ± 0.0 | 65.2 ± 0.5 | 1.3 ± 0.4 | 67.8 ± 0.8 | 1.4 ± 0.7 | **67.9 ± 0.9** | **1.4 ± 0.7** |
| Pokec-z | 0.00 | 68.4 ± 0.4 | 2.6 ± 0.7 | 68.4 ± 0.4 | 2.6 ± 0.7 | 68.4 ± 0.4 | 2.6 ± 0.7 | 68.4 ± 0.4 | 2.6 ± 0.7 | 68.4 ± 0.4 | 2.6 ± 0.7 |
| | 0.05 | **69.0 ± 0.4** | **3.4 ± 0.5** | 67.1 ± 0.5 | 2.7 ± 1.0 | 68.1 ± 0.4 | 2.9 ± 0.3 | 68.7 ± 0.5 | 2.9 ± 0.5 | 68.7 ± 0.4 | 3.1 ± 1.0 |
| | 0.10 | 68.7 ± 0.1 | 2.4 ± 0.5 | 66.3 ± 0.6 | 1.7 ± 0.6 | 68.2 ± 0.5 | 1.7 ± 0.5 | **69.0 ± 0.6** | **2.9 ± 0.6** | 69.0 ± 0.5 | 3.0 ± 0.6 |
| | 0.15 | 67.9 ± 0.3 | 2.8 ± 0.3 | 65.5 ± 0.3 | 1.4 ± 0.3 | 67.0 ± 0.5 | 1.3 ± 0.2 | 68.6 ± 0.5 | 2.9 ± 0.6 | **69.0 ± 0.7** | **2.7 ± 0.4** |
| | 0.20 | 67.9 ± 0.3 | 2.2 ± 0.6 | 64.2 ± 0.4 | 0.7 ± 0.3 | 66.1 ± 0.1 | 1.6 ± 0.5 | 68.8 ± 0.4 | 3.0 ± 0.4 | **69.2 ± 0.4** | **2.9 ± 0.3** |
| | 0.25 | 67.6 ± 0.3 | 1.9 ± 0.3 | 64.2 ± 0.3 | 0.5 ± 0.1 | 65.1 ± 0.3 | 1.9 ± 0.6 | 69.1 ± 0.3 | 2.9 ± 0.7 | **69.3 ± 0.3** | **2.7 ± 0.6** |
| Bail | 0.00 | 93.1 ± 0.2 | 7.2 ± 0.6 | 93.1 ± 0.2 | 7.2 ± 0.6 | 93.1 ± 0.2 | 7.2 ± 0.6 | 93.1 ± 0.2 | 7.2 ± 0.6 | 93.1 ± 0.2 | 7.2 ± 0.6 |
| | 0.05 | 92.1 ± 0.3 | 8.0 ± 1.9 | 91.8 ± 0.1 | 7.1 ± 1.1 | 91.2 ± 0.2 | 5.6 ± 0.7 | **93.0 ± 0.3** | **7.8 ± 1.0** | 92.9 ± 0.2 | 7.7 ± 1.0 |
| | 0.10 | 91.6 ± 0.1 | 7.3 ± 1.2 | 90.3 ± 0.1 | 6.1 ± 0.6 | 90.3 ± 0.1 | 5.1 ± 0.4 | **93.0 ± 0.1** | **8.0 ± 0.7** | 92.9 ± 0.2 | 7.9 ± 0.8 |
| | 0.15 | 91.3 ± 0.1 | 6.5 ± 0.9 | 89.4 ± 0.0 | 4.8 ± 0.1 | 89.8 ± 0.1 | 5.2 ± 0.1 | **93.1 ± 0.1** | **8.2 ± 0.6** | 93.0 ± 0.2 | 7.8 ± 0.8 |
| | 0.20 | 91.2 ± 0.2 | 6.6 ± 0.6 | 88.5 ± 0.1 | 4.0 ± 0.4 | 89.3 ± 0.1 | 5.3 ± 0.4 | **93.1 ± 0.1** | **7.9 ± 0.6** | 93.1 ± 0.1 | 8.2 ± 0.6 |
| | 0.25 | 90.9 ± 0.1 | 6.8 ± 0.8 | 87.4 ± 0.3 | 3.6 ± 0.5 | 88.9 ± 0.1 | 5.4 ± 0.3 | 92.9 ± 0.1 | 7.6 ± 0.5 | **93.0 ± 0.2** | **7.8 ± 0.7** |

the effectiveness of FATE with a different victim model, i.e., InFoRM-GNN [20], which mitigates individual bias.

# 7 Related Work

**Algorithmic fairness on graphs** aims to obtain debiased graph learning results such that a pre-defined fairness definition can be satisfied with respect to the nodes/edges in the graph. Several definitions of the fairness has been studied so far. Group fairness in graph embedding can be ensured via several ways, including adversarial learning-based methods [5, 11], random walk-based methods [36, 25] and dropout-based methods [39]. Individual fairness on graphs can be ensured via Lipschitz regularization [20] and learning-to-rank [13]. Other than the aforementioned two fairness definitions, several other fairness definitions are studied in the context of graph learning, including counterfactual fairness [1, 31], degree fairness [42, 24, 29], dyadic fairness [32, 27] and max-min fairness [37, 43]. For a comprehensive review of related works, please refer to existing surveys [50, 10, 14] and tutorials [22, 23]. It should be noted that our work aims to attack fairness (i.e., making the model more biased) rather than ensuring fairness as in the aforementioned literature.

**Adversarial attacks on graphs** aim to exacerbate the utility of graph learning models by perturbing the input graph topology and/or node features. Several approaches have been proposed to attack graph learning models, including reinforcement learning [12], bi-level optimization [51, 52], projected gradient descent [40, 48] and edge rewiring/flipping [4, 31]. Other than adversarial attacks that worsen the utility of a graph learning model, a few efforts have been made to attack the fairness of a machine learning model for IID tabular data via label flipping [33], adversarial data injection [38, 8], adversarial sampling [44]. Different from [38, 33, 8, 44], we aim to poison the input graph via structural modifications on the topology rather than injecting adversarial data sample(s). The most related work to our proposed method is by Hussain et al. [19], which degrade the group fairness of graph neural networks by randomly injecting edges for nodes in different demographic groups and with different class labels. In contrast, our proposed method could attack *any* fairness definition for *any* graph learning models via arbitrary edge manipulation operations, as long as the bias function and the utility loss are differentiable.

# 8 Conclusion

We study the problem of fairness attacks on graph learning models, whose goal is to amplify the bias while maintaining the utility on the downstream task. We formally define the problem as a bi-level optimization problem, where the upper-level optimization problem maximizes the bias function with respect to a user-defined fairness definition and the lower-level optimization problem minimizes a task-specific loss function. We then propose a meta learning-based framework named FATE to poison the input graph using the meta-gradient of the bias function with respect to the input graph. We instantiate FATE by attacking statistical parity on graph neural networks in a binary node classification problem with binary sensitive attributes. Empirical evaluation demonstrates that FATE is effective (consistently amplifying bias) and deceptive (achieving the highest micro F1 score).

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
