# Supplementary Material

## Organization of the Appendix

The supplementary material contains the following information.

- Appendix A provides additional examples of graph learning models from the optimization perspective.
- Appendix B presents the pseudocode of FATE.
- Appendix C offers the detailed parameter settings regarding the reproducibility of this paper.
- Appendix D provides additional experimental results on using FairGNN [11] and evaluating under macro F1 score and AUC score.
- Appendix E provides additional experimental results on using InFoRM-GNN [20] and evaluating under macro F1 score and AUC score.
- Appendix F shows the transferability of using FATE to attack the statistical parity and individual fairness of the non-convolutional aggregation-based graph attention network with linear GCN as the surrogate model.
- Appendix G discusses the relationship between fairness attacks and the impossibility theorem as well as Metattack [50].

Code can be found at the following repository:

https://anonymous.4open.science/r/FATE-BC55/README.md.

## A  Graph Learning Models from the Optimization Perspective

Here, we discuss four additional non-parameterized graph learning models from the optimization perspective, including PageRank, spectral clustering, matrix factorization-based completion and first-order LINE.

**Model #1: PageRank.** It is one of the most successful random walk based ranking algorithm to measure node importance. Mathematically, PageRank solves the linear system

$$\mathbf{r} = c\mathbf{P}\mathbf{r} + (1 - c)\mathbf{e} \tag{11}$$

where $c$ is the damping factor, $\mathbf{P}$ is the propagation matrix and $\mathbf{e}$ is the teleportation vector. In PageRank, the propagation matrix $\mathbf{P}$ is often defined as the row-normalized adjacency matrix of a graph $\mathcal{G}$ and the teleportation vector is a uniform distribution $\frac{1}{n}\mathbf{1}$ with $\mathbf{1}$ being a vector filled with $1$. Equivalently, given a damping factor $c$ and a teleportation vector $\mathbf{e}$, the PageRank vector $\mathbf{Y} = \mathbf{r}$ can be learned by minimizing the following loss function

$$\min_{\mathbf{r}} \quad c\mathbf{r}^T(\mathbf{I} - \mathbf{P})\mathbf{r} + (1 - c)\|\mathbf{r} - \mathbf{e}\|_2^2 \tag{12}$$

where $c\left(\mathbf{r}^T\left(\mathbf{I} - \mathbf{P}\right)\mathbf{r}\right)$ is a smoothness term and $(1 - c)\left\|\mathbf{r} - \mathbf{e}\right\|_2^2$ is a query-specific term. To attack the fairness of PageRank with FATE, the attacker could attack a surrogate PageRank with different choices of damping factor $c$ and/or teleportation vector $\mathbf{e}$.

**Model #2: Spectral clustering.** It aims to identify clusters of nodes such that the intra-cluster connectivity are maximized while inter-cluster connectivity are minimized. To find $k$ clusters of nodes, spectral clustering finds a soft cluster membership matrix $\mathbf{Y} = \mathbf{C}$ with orthonormal columns by minimizing the following loss function

$$\min_{\mathbf{C}} \quad \mathrm{Tr}\left(\mathbf{C}^T\mathbf{L}\mathbf{C}\right) \tag{13}$$

where $\mathbf{L}$ is the (normalized) graph Laplacian of the input graph $\mathcal{G}$. It is worth noting that the columns of learning result $\mathbf{C}$ is equivalent to the eigenvectors of $\mathbf{L}$ associated with smallest $k$ eigenvalues. To attack the fairness of spectral clustering with FATE, the attacker might attack a surrogate spectral clustering with different number of clusters $k$.

**Model #3: Matrix factorization-based completion.** Suppose we have a bipartite graph $\mathcal{G}$ with $n_1$ users, $n_2$ items and $m$ interactions between users and items. Matrix factorization-based completion aims to learn two low-rank matrices an $n_1 \times z$ matrix $\mathbf{U}$ and an $n_2 \times z$ matrix $\mathbf{V}$ such that the following loss function will be minimized

$$\min_{\mathbf{U},\mathbf{V}} \quad \| \operatorname{proj}_\Omega \left( \mathbf{R} - \mathbf{U}\mathbf{V}^T \right) \|_F^2 + \lambda_1 \|\mathbf{U}\|_F^2 \lambda_2 + \|\mathbf{V}\|_F^2 \tag{14}$$

where $\mathbf{A} = \begin{pmatrix} \mathbf{0}_{n_1} & \mathbf{R} \\ \mathbf{R}^T & \mathbf{0}_{n_2} \end{pmatrix}$ with $\mathbf{0}_{n_1}$ being an $n_1 \times n_1$ square matrix filled with 0, $\Omega = \{(i,j)|(i,j) \text{ is observed}\}$ is the set of observed interaction between any user $i$ and any item $j$, $\operatorname{proj}_\Omega (\mathbf{Z}) [i,j]$ equals to $\mathbf{Z}[i,j]$ if $(i,j) \in \Omega$ and 0 otherwise, $\lambda_1$ and $\lambda_2$ are two hyperparameters for regularization. To attack the fairness of matrix factorization-based completion with FATE, the attacker could attack a surrogate model with different number of latent factors $z$.

**Model #4: First-order LINE.** It is a skip-gram based node embedding model. The key idea of first-order LINE is to map each node into a $h$-dimensional space such that the dot product of the embeddings of any two connected nodes will be small. To achieve this goal, first-order LINE essentially optimizes the following loss function

$$\max_{\mathbf{H}} \sum_{i=1}^n \sum_{j=1}^n \mathbf{A}[i,j] \left( \log g \left( \mathbf{H}[j,:]\mathbf{H}[i,:]^T \right) + k \mathbb{E}_{j' \sim P_n} [\log g \left( -\mathbf{H}[j',:]\mathbf{H}[i,:]^T \right)] \right) \tag{15}$$

where $\mathbf{H}$ is the embedding matrix with $\mathbf{H}[i,:]$ being the $h$-dimensional embedding of node $i$, $g(x) = 1/(1 + e^{-x})$ is the sigmoid function, $k$ is the number of negative samples and $P_n$ is the distribution for negative sampling such that the sampling probability for node $i$ is proportional to its degree $\deg_i$. For a victim first-order LINE, the attacker could attack a surrogate LINE (1st) with different dimension $h$ in the embedding space and/or a different number of negative samples $g$.

**Remarks.** Note that, for a non-parameterized graph learning model (e.g., PageRank, spectral clustering, matrix completion, first-order LINE), we have $\Theta = \{\mathbf{Y}\}$ which is the set of learning results. For example, we have $\Theta = \{\mathbf{r}\}$ for PageRank, $\Theta = \{\mathbf{C}\}$ for spectral clustering, $\Theta = \{\mathbf{U}, \mathbf{V}\}$ and $\Theta = \{\mathbf{H}\}$ for LINE (1st). For parameterized graph learning models (e.g., GCN), $\Theta$ refers to the set of learnable weights, e.g., $\Theta = \{\mathbf{W}^{(1)}, \ldots, \mathbf{W}^{(L)}\}$ for an $L$-layer GCN.

# B Pseudocode of FATE

Algorithm 1 summarizes the detailed steps on fairness attack with FATE. To be specific, after initialization (line 1), we pre-train the surrogate graph learning model (lines $4 - 6$) and get the pre-trained surrogate model $\Theta^{(T)}$ as well as learning results $\mathbf{Y}^{(T)}$ (line 7). After that, we compute the meta gradient of the bias function (lines $8 - 11$) and perform either discretized attack or continuous attack based on the interest of attacker (i.e., discretized poisoning attack in lines $12 - 15$ or continuous poisoning attack in lines $16 - 18$).

# C Experimental Settings

In this section, we provide detailed information about the experimental settings. These include the hardware and software specifications, dataset descriptions, descriptions of baseline methods, evaluation metrics as well as detailed parameter settings.

## C.1 Hardware and Software Specifications

All codes are programmed in Python 3.8.13 and PyTorch 1.12.1. All experiments are performed on a Linux server with 96 Intel Xeon Gold 6240R CPUs and 4 Nvidia Tesla V100 SXM2 GPUs with 32 GB memory.

**Algorithm 1:** FATE

**Given :** an undirected graph $\mathcal{G} = \{\mathbf{A}, \mathbf{X}\}$, the set of training nodes $\mathcal{V}_{\text{train}}$, fairness-related auxiliary information matrix $\mathbf{F}$, total budget $B$, budget in step $i$ $\delta_i$, the bias function $b$, number of pre-training epochs $T$;

**Find :** the poisoned graph $\widetilde{\mathcal{G}}$;

1 poisoned graph $\widetilde{\mathcal{G}} \leftarrow \mathcal{G}$, cumulative budget $\Delta \leftarrow 0$, step counter $i \leftarrow 0$;
2 **while** $\Delta < B$ **do**
3     $\nabla_{\widetilde{\mathcal{G}}} b \leftarrow 0$;
4     **for** $t = 1$ *to* $T$ **do**
5        update $\Theta^{(t)}$ to $\Theta^{(t+1)}$ with a gradient-based optimizer (e.g., Adam);
6     **end**
7     get $\mathbf{Y}^{(T)}$ and $\Theta^{(T)}$;
8     compute meta-gradient $\nabla_{\mathcal{G}} b \leftarrow \nabla_{\Theta^{(T)}} b \left(\mathbf{Y}, \Theta^{(T)}, \mathbf{F}\right) \cdot \nabla_{\mathcal{G}} \Theta^{(T)}$;
9     **if** *attack the adjacency matrix* **then**
10        compute the derivative $\nabla_{\widetilde{\mathbf{A}}} b \leftarrow \nabla_{\widetilde{\mathbf{A}}} b + \left(\nabla_{\widetilde{\mathbf{A}}} b\right)^T - \text{diag}\left(\nabla_{\widetilde{\mathbf{A}}} b\right)$;
11     **end**
12     **if** *discretized poisoning attack* **then**
13        compute the poisoning preference matrix $\nabla_{\widetilde{\mathbf{A}}}$ by Eq. (7);
14        select the edges to poison in $\nabla_{\widetilde{\mathbf{A}}}$ with budget $\delta_i$ by Eq. (8);
15        update the corresponding entries in $\widetilde{\mathcal{G}}$;
16     **else**
17        update $\widetilde{\mathcal{G}}$ by Eq. (6) with budget $\delta_i$;
18     **end**
19     $\Delta \leftarrow \Delta + \delta_i$;
20     $i \leftarrow i + 1$;
21 **end**
22 **return** $\widetilde{\mathcal{G}}$;

## C.2 Dataset Descriptions

We use three widely-used benchmark datasets for fair graph learning: **Pokec-z**, **Pokec-n** and **Bail**. For each dataset, we use a fixed random seed to split the dataset into training, validation and test sets with the split ratio being 50%, 25%, and 25%, respectively. The statistics of the datasets, including the number of nodes (# Nodes), the number of edges (# Edges), the number of features (# Features), the sensitive attribute (Sensitive Attr.) and the label (Label), are summarized in Table 3.

- **Pokec-z** and **Pokec-n** are two datasets collected from the Slovakian social network *Pokec*, each of which represents a sub-network of a province. Each node in these datasets is a user belonging to two major regions of the corresponding provinces, and each edge is the friendship relationship between two users. The sensitive attribute is the user region, and the label is the working field of a user.

- **Bail** is a similarity graph of criminal defendants during $1990 - 2009$. Each node is a defendant during this time period. Two nodes are connected if they share similar past criminal records and demographics. The sensitive attribute is the race of the defendant, and the label is whether the defendant is on bail or not.

Table 3: Statistics of the datasets.

| Dataset | Pokec-z | Pokec-n | Bail |
|---|---|---|---|
| # Nodes | $7,659$ | $6,185$ | $18,876$ |
| # Edges | $20,550$ | $15,321$ | $311,870$ |
| # Features | $276$ | $265$ | $17$ |
| Sensitive Attr. | Region | Region | Race |
| Label | Working field | Working field | Bail decision |

### C.3 Descriptions of Baseline Methods

We compare the proposed FATE with baseline methods, including **Random**, **DICE** [44] and **FA-GNN** [19]. Descriptions of the baseline methods are as follows.

- **Random** is a heuristic approach that randomly inserts edges to the input graph.
- **DICE** [44] attacks the utility of a graph learning algorithm by randomly deleting edges within a community and inserting edges across different communities. Similar to [49, 19], the community is defined as the group of nodes with the same class label.
- **FA-GNN** [19] aims to attack the fairness of a graph neural network by adversarially inserting edges that connect nodes in consideration of their class labels and sensitive attribute values.

### C.4 Evaluation Metrics

In our experiments, we aim to evaluate how effective FATE is in (1) attacking the fairness and (2) maintaining the utility of node classification.

To evaluate the performance of FATE in attacking the group fairness, we evaluate the effectiveness using $\Delta_{\text{SP}}$, which is defined as follows.

$$\Delta_{\text{SP}} = |P(\widehat{y} = 1 \mid s = 1) - P(\widehat{y} = 1 \mid s = 0)| \tag{16}$$

where $s$ is the sensitive attribute value of a node and $\widehat{y}$ is the ground-truth and predicted class labels of a node. While to evaluate the performance of FATE in attacking the individual fairness, we evaluate the effectiveness using the InFoRM bias (Bias) measure [20], which is defined as follows.

$$\text{Bias} = \sum_{i \in \mathcal{V}_{\text{test}}} \sum_{j \in \mathcal{V}_{\text{test}}} \mathbf{S}[i,j] \|\mathbf{Y}[i,:] - \mathbf{Y}[j,:]\|_F^2 \tag{17}$$

where $\mathcal{V}_{\text{test}}$ is the set of test nodes and $\mathbf{S}$ is the oracle pairwise node similarity matrix. The intuition of Eq. (17) is to measure the squared difference between the learning results of two test nodes, weighted by their pairwise similarity.

To evaluate the performance of FATE in maintaining the utility, we use micro F1 score (Micro F1), macro F1 score (Macro F1) and AUC score.

### C.5 Detailed Parameter Settings

**Poisoning the input graph.** During poisoning attacks, we set a fixed random seed to control the randomness. The random seed used for each dataset in attacking group/individual fairness are summarized in Table 4.

- **Surrogate model training.** We run all methods with a perturbation rate from 0.05 to 0.25 with a step size of 0.05. For FA-GNN [19], we follow its official implementation and use the same surrogate 2-layer GCN [26] with 16 hidden dimensions for poisoning attack.[4] The surrogate GCN in FA-GNN is trained for 500 epochs with a learning rate $1e-2$, weight decay $5e-4$, and dropout rate 0.5. For FATE, we use a 2-layer linear GCN [45] with 16 hidden dimensions for poisoning attacks. And the surrogate linear GCN in FATE is trained for 500 epochs with a learning rate $1e-2$, weight decay $5e-4$, and dropout rate 0.5.

- **Graph topology manipulation.** For Random and DICE, we use the implementations provided in the deeprobust package with the default parameters to add the adversarial edges.[5] For FA-GNN, we add adversarial edges that connect two nodes with different class labels and different sensitive attributes, which provides the most promising performance as shown in [19]. For FATE, suppose we poison the input graph in $p$ ($p > 1$) attacking steps. Then the per-iteration attacking budget in Algorithm 1 is set as $\delta_1 = 1$ and $\delta_i = \frac{r|\mathcal{E}|-1}{p-1}$, $\forall i \in \{2, \dots, p\}$, where $r$ is the perturbation rate and $|\mathcal{E}|$ is the number of edges. Detailed choices of $p$ for each dataset in attacking group/individual fairness are summarized in Table 4.

---

[4] https://github.com/mengcao327/attack-gnn-fairness
[5] https://deeprobust.readthedocs.io/

Table 4: Parameter settings on the random seed for all baseline methods in poisoning attacks (Random Seed) and the number of steps for poisoning attacks in FATE (Attacking Steps).

| Dataset | Fairness Definition | Attacking Steps | Random Seed |
|---------|---------------------|-----------------|-------------|
| Pokec-n | Statistical parity | 3 | 25 |
|         | Individual fairness | 3 | 45 |
| Pokec-z | Statistical parity | 3 | 25 |
|         | Individual fairness | 5 | 15 |
| Bail    | Statistical parity | 3 | 25 |
|         | Individual fairness | 3 | 5 |

**Training the victim model.** We use a fixed list of random seed ($[0, 1, 2, 42, 100]$) to train each victim model 5 times and report the mean and standard deviation. Regarding the victim models in group fairness attacks, we train a 2-layer GCN [26] for 400 epochs and a 2-layer FairGNN [11] for 2000 epochs to evaluate the efficacy of fairness attacks. The hidden dimension, learning rate, weight decay and dropout rate of GCN and FairGNN are set to 128, $1e-3$, $1e-5$ and 0.5, respectively. The regularization parameters in FairGNN, namely $\alpha$ and $\beta$, are set to 100 and 1 for all datasets, respectively. Regarding the victim models in individual fairness attacks, we train a 2-layer GCN [26] and 2-layer InFoRM-GNN [20, 13] for 400 epochs. The hidden dimension, learning rate, weight decay and dropout rate of GCN and InFoRM-GNN are set to 128, $1e-3$, $1e-5$ and 0.5, respectively. The regularization parameter in InFoRM-GNN is set to 0.1 for all datasets.

# D   Additional Experimental Results: Attacking Statistical Parity on Graph Neural Networks

**A – FATE with FairGNN as the victim model.** Here, we study how robust FairGNN is in fairness attacks against statistical parity with linear GCN as the surrogate model. Note that FairGNN is a fairness-aware graph neural network that leverages adversarial learning to ensure statistical parity.

**Main results.** Similar to Section 6.1, for FATE, we conduct fairness attacks via both edge flipping (FATE-flip in Table 5 and edge addition (FATE-add in Table 5). For all other baseline methods, edges are only added. From Table 5, we have the following key observations: (1) Even though the surrogate model is linear GCN without fairness consideration, FairGNN, which ensures statistical parity on graph neural networks, cannot mitigate the bias caused by fairness attacks and is vulnerable to fairness attack. (2) FATE-flip and FATE-add are effective and the most deceptive method in fairness attacks. (3) FATE-flip and FATE-add are the only methods that consistently succeed in fairness attacks, while all other baseline methods might fail in some cases (indicated by the underlined $\Delta_{SP}$ in both tables). In short, even when the victim model is FairGNN (a fair graph neural network), our proposed FATE framework are effective in fairness attacks while being the most deceptive (i.e., highest micro F1 score).

**Effect of the perturbation rate.** From Table 5, we can find out that: (1) $\Delta_{SP}$ tends to increase when the perturbation rate increases, indicating the effectiveness of FATE-flip and FATE-add for attacking fairness. (2) There is no clear correlation between the perturbation rate and the micro F1 scores of FATE-flip and FATE-add, meaning that they are deceptive in maintaining the utility. As a consequence, FATE is effective and deceptive in attacking fairness of FairGNN across different perturbation rates.

**B – Performance evaluation under different utility metrics.** Here we provide additional evaluation results of utility using macro F1 score and AUC score. From Tables 6 and 7, we can see that macro F1 scores and AUC scores are less impacted by different perturbation rates. Thus, it provide additional evidence that our proposed FATE framework can achieve deceptive fairness attacks by achieving comparable or even better utility on the semi-supervised node classification.

Table 5: Effectiveness of attacking group fairness on FairGNN. FATE poisons the graph via both edge flipping (FATE-flip) and edge addition (FATE-add) while all other baselines poison the graph via edge addition. Higher is better (↑) for micro F1 score (Micro F1) and $\Delta_{\mathrm{SP}}$. Bold font indicates the success of fairness attack (i.e., bias is increased after attack) with the highest micro F1 score. Underlined cell indicates the failure of fairness attack (i.e., $\Delta_{\mathrm{SP}}$ is decreased after attack).

| Dataset | Ptb. | Random | | DICE | | FA-GNN | | FATE-flip | | FATE-add | |
|---|---|---|---|---|---|---|---|---|---|---|---|
| | | Micro F1 (↑) | $\Delta_{\mathrm{SP}}$ (↑) | Micro F1 (↑) | $\Delta_{\mathrm{SP}}$ (↑) | Micro F1 (↑) | $\Delta_{\mathrm{SP}}$ (↑) | Micro F1 (↑) | $\Delta_{\mathrm{SP}}$ (↑) | Micro F1 (↑) | $\Delta_{\mathrm{SP}}$ (↑) |
| Pokec-n | 0.00 | 68.2 ± 0.4 | 6.7 ± 2.0 | 68.2 ± 0.4 | 6.7 ± 2.0 | 68.2 ± 0.4 | 6.7 ± 2.0 | 68.2 ± 0.4 | 6.7 ± 2.0 | 68.2 ± 0.4 | 6.7 ± 2.0 |
| | 0.05 | 67.4 ± 0.8 | 8.2 ± 2.5 | 66.2 ± 0.6 | 9.9 ± 1.7 | 66.7 ± 1.2 | 2.8 ± 1.3 | 68.4 ± 0.2 | 8.9 ± 1.8 | 68.4 ± 0.2 | 8.9 ± 1.8 |
| | 0.10 | 67.5 ± 0.5 | 8.3 ± 1.5 | 66.3 ± 0.4 | 9.5 ± 2.1 | 66.6 ± 0.5 | 5.9 ± 1.3 | 68.5 ± 0.4 | 9.5 ± 1.4 | 68.5 ± 0.4 | 9.5 ± 1.4 |
| | 0.15 | 65.9 ± 0.6 | 10.4 ± 2.3 | 65.4 ± 0.6 | 9.2 ± 2.6 | 64.8 ± 1.6 | 9.0 ± 3.3 | 68.5 ± 0.8 | 10.5 ± 2.6 | 68.5 ± 0.8 | 10.5 ± 2.6 |
| | 0.20 | 65.4 ± 0.5 | 10.0 ± 1.5 | 65.0 ± 0.4 | 4.4 ± 2.5 | 65.2 ± 0.2 | 11.6 ± 2.6 | 68.3 ± 0.3 | 10.7 ± 2.3 | 68.3 ± 0.3 | 10.7 ± 2.3 |
| | 0.25 | 65.8 ± 1.1 | 7.5 ± 1.9 | 63.8 ± 0.3 | 5.4 ± 1.8 | 64.8 ± 0.8 | 14.2 ± 2.3 | 68.5 ± 0.3 | 9.1 ± 3.6 | 68.5 ± 0.3 | 9.1 ± 3.6 |
| Pokec-z | 0.00 | 68.7 ± 0.3 | 7.0 ± 0.9 | 68.7 ± 0.3 | 7.0 ± 0.9 | 68.7 ± 0.3 | 7.0 ± 0.9 | 68.7 ± 0.3 | 7.0 ± 0.9 | 68.7 ± 0.3 | 7.0 ± 0.9 |
| | 0.05 | 67.3 ± 0.6 | 8.7 ± 2.8 | 67.5 ± 0.4 | 8.5 ± 1.3 | 67.1 ± 1.0 | 1.7 ± 1.3 | 68.7 ± 0.4 | 8.0 ± 0.9 | 68.7 ± 0.4 | 8.0 ± 0.9 |
| | 0.10 | 67.1 ± 0.2 | 8.6 ± 2.7 | 66.1 ± 0.3 | 7.0 ± 2.9 | 65.9 ± 0.8 | 6.8 ± 1.7 | 68.5 ± 0.5 | 9.0 ± 1.8 | 68.5 ± 0.5 | 9.0 ± 1.8 |
| | 0.15 | 66.8 ± 0.8 | 8.9 ± 2.2 | 65.2 ± 0.5 | 6.6 ± 1.4 | 64.9 ± 0.9 | 10.0 ± 1.7 | 68.7 ± 0.5 | 9.5 ± 2.2 | 68.7 ± 0.5 | 9.5 ± 2.2 |
| | 0.20 | 66.8 ± 0.7 | 8.6 ± 3.0 | 63.7 ± 0.6 | 6.6 ± 2.9 | 64.6 ± 0.8 | 14.2 ± 3.1 | 68.8 ± 0.2 | 10.4 ± 1.6 | 68.8 ± 0.2 | 10.4 ± 1.6 |
| | 0.25 | 66.4 ± 0.4 | 7.9 ± 2.8 | 63.4 ± 0.4 | 6.0 ± 2.8 | 64.0 ± 1.1 | 14.0 ± 2.0 | 68.5 ± 0.3 | 10.3 ± 2.1 | 68.5 ± 0.3 | 10.3 ± 2.1 |
| Bail | 0.00 | 93.9 ± 0.1 | 8.4 ± 0.2 | 93.9 ± 0.1 | 8.4 ± 0.2 | 93.9 ± 0.1 | 8.4 ± 0.2 | 93.9 ± 0.1 | 8.4 ± 0.2 | 93.9 ± 0.1 | 8.4 ± 0.2 |
| | 0.05 | 90.6 ± 1.2 | 8.3 ± 0.2 | 89.1 ± 1.2 | 8.3 ± 0.3 | 89.1 ± 2.0 | 10.8 ± 1.1 | 93.6 ± 0.1 | 9.2 ± 0.2 | 93.6 ± 0.1 | 9.1 ± 0.2 |
| | 0.10 | 90.1 ± 2.0 | 8.5 ± 0.6 | 88.1 ± 1.8 | 8.2 ± 0.3 | 87.3 ± 2.2 | 12.2 ± 1.2 | 93.4 ± 0.1 | 9.3 ± 0.2 | 93.4 ± 0.1 | 9.3 ± 0.2 |
| | 0.15 | 90.0 ± 2.0 | 8.1 ± 0.5 | 86.9 ± 2.0 | 8.1 ± 0.5 | 87.8 ± 2.0 | 10.9 ± 2.1 | 93.3 ± 0.1 | 9.2 ± 0.3 | 93.3 ± 0.1 | 9.2 ± 0.3 |
| | 0.20 | 89.2 ± 2.4 | 8.4 ± 0.7 | 85.3 ± 2.7 | 8.2 ± 0.4 | 86.0 ± 2.7 | 11.7 ± 2.4 | 93.1 ± 0.2 | 9.3 ± 0.3 | 93.0 ± 0.1 | 9.4 ± 0.2 |
| | 0.25 | 88.8 ± 2.3 | 8.2 ± 0.7 | 85.3 ± 3.3 | 7.9 ± 0.5 | 87.0 ± 1.9 | 8.5 ± 2.6 | 93.0 ± 0.1 | 9.2 ± 0.4 | 93.0 ± 0.2 | 9.3 ± 0.3 |

Table 6: Macro F1 score and AUC score of attacking group fairness on GCN. FATE poisons the graph via both edge flipping (FATE-flip) and edge addition (FATE-add) while all other baselines poison the graph via edge addition. Higher is better (↑) for macro F1 score (Macro F1) and AUC score (AUC). Bold font indicates the highest macro F1 score or AUC score.

| Dataset | Ptb. | Random | | DICE | | FA-GNN | | FATE-flip | | FATE-add | |
|---|---|---|---|---|---|---|---|---|---|---|---|
| | | Macro F1 (↑) | AUC (↑) | Macro F1 (↑) | AUC (↑) | Macro F1 (↑) | AUC (↑) | Macro F1 (↑) | AUC (↑) | Macro F1 (↑) | AUC (↑) |
| Pokec-n | 0.00 | 65.3 ± 0.3 | 69.9 ± 0.5 | 65.3 ± 0.3 | 69.9 ± 0.5 | 65.3 ± 0.3 | 69.9 ± 0.5 | 65.3 ± 0.3 | 69.9 ± 0.5 | 65.3 ± 0.3 | 69.9 ± 0.5 |
| | 0.05 | 65.7 ± 0.3 | 70.4 ± 0.4 | 64.9 ± 0.2 | 69.8 ± 0.3 | 64.9 ± 0.2 | 70.4 ± 0.2 | 66.0 ± 0.3 | 70.3 ± 0.6 | 66.0 ± 0.3 | 70.3 ± 0.6 |
| | 0.10 | 64.6 ± 0.4 | 69.6 ± 0.3 | 63.4 ± 0.6 | 67.7 ± 0.3 | 64.1 ± 0.3 | 70.0 ± 0.1 | 66.1 ± 0.6 | 70.4 ± 0.6 | 66.1 ± 0.6 | 70.4 ± 0.6 |
| | 0.15 | 65.1 ± 0.4 | 69.6 ± 0.1 | 62.8 ± 0.7 | 67.3 ± 0.4 | 64.3 ± 0.6 | 69.1 ± 0.5 | 66.1 ± 0.2 | 70.6 ± 0.6 | 66.1 ± 0.2 | 70.6 ± 0.6 |
| | 0.20 | 64.5 ± 0.5 | 69.1 ± 0.1 | 60.9 ± 0.2 | 64.6 ± 0.2 | 63.5 ± 0.2 | 68.0 ± 0.2 | 66.4 ± 0.3 | 70.7 ± 0.4 | 66.4 ± 0.3 | 70.7 ± 0.4 |
| | 0.25 | 64.5 ± 0.6 | 68.8 ± 0.1 | 59.7 ± 0.6 | 63.6 ± 0.6 | 63.5 ± 0.3 | 69.5 ± 0.3 | 66.3 ± 0.3 | 70.6 ± 0.6 | 66.3 ± 0.3 | 70.6 ± 0.6 |
| Pokec-z | 0.00 | 68.2 ± 0.4 | 75.1 ± 0.3 | 68.2 ± 0.4 | 75.1 ± 0.3 | 68.2 ± 0.4 | 75.1 ± 0.3 | 68.2 ± 0.4 | 75.1 ± 0.3 | 68.2 ± 0.4 | 75.1 ± 0.3 |
| | 0.05 | 68.5 ± 0.4 | 74.5 ± 0.4 | 67.2 ± 0.5 | 74.2 ± 0.3 | 67.9 ± 0.3 | 74.5 ± 0.2 | 68.6 ± 0.4 | 75.2 ± 0.4 | 68.6 ± 0.4 | 75.2 ± 0.4 |
| | 0.10 | 68.5 ± 0.3 | 74.8 ± 0.3 | 66.3 ± 0.2 | 72.9 ± 0.1 | 67.5 ± 0.5 | 73.8 ± 0.3 | 68.6 ± 0.6 | 75.2 ± 0.3 | 68.6 ± 0.6 | 75.2 ± 0.3 |
| | 0.15 | 67.8 ± 0.3 | 74.4 ± 0.3 | 65.3 ± 1.0 | 70.0 ± 0.8 | 66.1 ± 0.6 | 72.7 ± 0.2 | 68.9 ± 0.7 | 75.3 ± 0.2 | 68.9 ± 0.7 | 75.3 ± 0.2 |
| | 0.20 | 68.2 ± 0.4 | 74.5 ± 0.6 | 62.6 ± 0.6 | 68.0 ± 0.9 | 66.1 ± 0.2 | 71.9 ± 0.1 | 68.4 ± 0.5 | 75.1 ± 0.3 | 68.4 ± 0.5 | 75.1 ± 0.3 |
| | 0.25 | 68.0 ± 0.4 | 74.0 ± 0.4 | 63.4 ± 0.4 | 68.6 ± 0.6 | 65.3 ± 0.6 | 71.2 ± 0.3 | 68.4 ± 1.1 | 74.4 ± 1.4 | 68.4 ± 1.1 | 74.4 ± 1.4 |
| Bail | 0.00 | 92.3 ± 0.2 | 97.4 ± 0.1 | 92.3 ± 0.2 | 97.4 ± 0.1 | 92.3 ± 0.2 | 97.4 ± 0.1 | 92.3 ± 0.2 | 97.4 ± 0.1 | 92.3 ± 0.2 | 97.4 ± 0.1 |
| | 0.05 | 92.0 ± 0.2 | 95.3 ± 0.2 | 90.6 ± 0.3 | 93.8 ± 0.2 | 90.8 ± 0.1 | 94.4 ± 0.2 | 91.8 ± 0.1 | 97.1 ± 0.1 | 91.7 ± 0.1 | 97.1 ± 0.2 |
| | 0.10 | 91.4 ± 0.2 | 94.7 ± 0.3 | 89.2 ± 0.1 | 92.2 ± 0.3 | 89.5 ± 0.1 | 93.5 ± 0.1 | 91.6 ± 0.2 | 96.9 ± 0.1 | 91.6 ± 0.2 | 96.9 ± 0.1 |
| | 0.15 | 91.1 ± 0.2 | 94.2 ± 0.2 | 87.8 ± 0.2 | 91.1 ± 0.2 | 88.7 ± 0.3 | 92.5 ± 0.2 | 91.4 ± 0.2 | 96.9 ± 0.1 | 91.5 ± 0.1 | 96.9 ± 0.1 |
| | 0.20 | 90.7 ± 0.2 | 94.1 ± 0.1 | 86.9 ± 0.1 | 90.2 ± 0.2 | 88.4 ± 0.1 | 92.2 ± 0.1 | 91.3 ± 0.2 | 96.8 ± 0.1 | 91.4 ± 0.2 | 96.8 ± 0.1 |
| | 0.25 | 90.4 ± 0.2 | 93.4 ± 0.3 | 86.2 ± 0.1 | 89.2 ± 0.3 | 88.5 ± 0.2 | 92.0 ± 0.1 | 91.2 ± 0.1 | 96.8 ± 0.1 | 91.3 ± 0.2 | 96.8 ± 0.1 |

Table 7: Macro F1 score and AUC score of attacking group fairness on FairGNN. FATE poisons the graph via both edge flipping (FATE-flip) and edge addition (FATE-add) while all other baselines poison the graph via edge addition. Higher is better (↑) for macro F1 score (Macro F1) and AUC score (AUC). Bold font indicates the highest macro F1 score or AUC score.

| Dataset | Ptb. | Random | | DICE | | FA-GNN | | FATE-flip | | FATE-add | |
|---|---|---|---|---|---|---|---|---|---|---|---|
| | | Macro F1 (↑) | AUC (↑) | Macro F1 (↑) | AUC (↑) | Macro F1 (↑) | AUC (↑) | Macro F1 (↑) | AUC (↑) | Macro F1 (↑) | AUC (↑) |
| Pokec-n | 0.00 | 65.6 ± 0.3 | 70.4 ± 0.5 | 65.6 ± 0.3 | 70.4 ± 0.5 | 65.6 ± 0.3 | 70.4 ± 0.5 | 65.6 ± 0.3 | 70.4 ± 0.5 | 65.6 ± 0.3 | 70.4 ± 0.5 |
| | 0.05 | 64.3 ± 0.6 | 68.3 ± 1.1 | 64.2 ± 0.8 | 68.4 ± 1.4 | 63.6 ± 0.7 | 68.2 ± 0.5 | 65.8 ± 0.3 | 70.7 ± 0.4 | 65.8 ± 0.3 | 70.7 ± 0.4 |
| | 0.10 | 63.8 ± 0.2 | 67.3 ± 1.1 | 63.4 ± 0.8 | 67.4 ± 0.6 | 63.9 ± 0.4 | 68.3 ± 0.2 | 66.0 ± 0.7 | 70.8 ± 0.5 | 66.0 ± 0.7 | 70.8 ± 0.5 |
| | 0.15 | 63.5 ± 0.2 | 67.8 ± 0.4 | 60.7 ± 0.7 | 65.2 ± 1.2 | 63.1 ± 0.6 | 67.2 ± 0.5 | 65.8 ± 1.0 | 70.8 ± 0.5 | 65.8 ± 1.0 | 70.8 ± 0.5 |
| | 0.20 | 63.1 ± 0.6 | 67.8 ± 1.1 | 60.2 ± 1.0 | 63.7 ± 1.1 | 62.3 ± 0.6 | 66.7 ± 0.9 | 65.7 ± 0.7 | 70.4 ± 0.5 | 65.7 ± 0.7 | 70.4 ± 0.5 |
| | 0.25 | 62.4 ± 0.3 | 66.8 ± 0.8 | 57.3 ± 0.4 | 61.8 ± 0.7 | 62.4 ± 1.4 | 67.6 ± 1.3 | 65.1 ± 1.2 | 70.1 ± 0.5 | 65.1 ± 1.2 | 70.1 ± 0.5 |
| Pokec-z | 0.00 | 68.4 ± 0.4 | 75.1 ± 0.3 | 68.4 ± 0.4 | 75.1 ± 0.3 | 68.4 ± 0.4 | 75.1 ± 0.3 | 68.4 ± 0.4 | 75.1 ± 0.3 | 68.4 ± 0.4 | 75.1 ± 0.3 |
| | 0.05 | 66.3 ± 0.9 | 73.5 ± 0.9 | 66.4 ± 0.8 | 72.8 ± 1.2 | 66.5 ± 1.4 | 72.6 ± 1.4 | 68.4 ± 0.4 | 74.7 ± 0.9 | 68.4 ± 0.4 | 74.7 ± 0.9 |
| | 0.10 | 66.0 ± 0.7 | 72.9 ± 1.1 | 64.9 ± 0.8 | 71.4 ± 0.4 | 65.2 ± 0.9 | 71.3 ± 1.7 | 68.2 ± 0.8 | 75.3 ± 0.8 | 68.2 ± 0.8 | 75.3 ± 0.8 |
| | 0.15 | 66.0 ± 0.8 | 71.8 ± 2.1 | 63.9 ± 0.8 | 68.5 ± 1.5 | 63.4 ± 1.5 | 70.0 ± 1.8 | 68.3 ± 0.5 | 75.2 ± 0.6 | 68.3 ± 0.5 | 75.2 ± 0.6 |
| | 0.20 | 65.6 ± 0.9 | 71.9 ± 1.4 | 62.1 ± 1.1 | 67.9 ± 1.8 | 63.7 ± 0.9 | 68.9 ± 1.6 | 68.3 ± 0.3 | 75.5 ± 0.3 | 68.3 ± 0.3 | 75.5 ± 0.3 |
| | 0.25 | 65.0 ± 0.7 | 71.2 ± 1.7 | 61.9 ± 0.4 | 66.7 ± 1.1 | 62.8 ± 1.8 | 69.4 ± 1.5 | 68.0 ± 0.5 | 75.3 ± 0.3 | 68.0 ± 0.5 | 75.3 ± 0.3 |
| Bail | 0.00 | 93.3 ± 0.2 | 97.4 ± 0.1 | 93.3 ± 0.2 | 97.4 ± 0.1 | 93.3 ± 0.2 | 97.4 ± 0.1 | 93.3 ± 0.2 | 97.4 ± 0.1 | 93.3 ± 0.2 | 97.4 ± 0.1 |
| | 0.05 | 89.5 ± 1.5 | 92.8 ± 1.8 | 87.8 ± 1.3 | 90.9 ± 1.5 | 87.8 ± 2.2 | 91.2 ± 1.8 | 93.0 ± 0.1 | 97.3 ± 0.1 | 93.0 ± 0.1 | 97.3 ± 0.1 |
| | 0.10 | 89.1 ± 2.2 | 92.7 ± 2.5 | 86.7 ± 2.0 | 90.1 ± 1.0 | 85.6 ± 2.7 | 90.5 ± 1.7 | 92.7 ± 0.1 | 97.1 ± 0.1 | 92.7 ± 0.1 | 97.1 ± 0.1 |
| | 0.15 | 88.8 ± 2.2 | 92.4 ± 2.5 | 85.2 ± 2.3 | 88.0 ± 2.8 | 86.1 ± 2.4 | 90.3 ± 2.2 | 92.6 ± 0.1 | 97.0 ± 0.1 | 92.6 ± 0.1 | 97.0 ± 0.1 |
| | 0.20 | 87.8 ± 2.8 | 91.6 ± 2.5 | 83.2 ± 3.1 | 86.5 ± 3.6 | 84.1 ± 3.0 | 89.0 ± 1.5 | 92.5 ± 0.2 | 97.0 ± 0.1 | 92.3 ± 0.1 | 97.0 ± 0.1 |
| | 0.25 | 87.5 ± 2.6 | 91.5 ± 2.6 | 83.3 ± 3.9 | 87.3 ± 3.7 | 85.1 ± 2.3 | 89.6 ± 1.3 | 92.3 ± 0.1 | 97.0 ± 0.1 | 92.3 ± 0.2 | 97.0 ± 0.1 |

# E   Additional Experimental Results: Attacking Individual Fairness on Graph Neural Networks

**A – FATE with InFoRM-GNN as the victim model.** InFoRM-GNN is an individually fair graph neural network that ensures individual fairness through regularizing the individual bias measure defined in Section 5. Here, we study how robust InFoRM-GNN is in fairness attacks against individual fairness with linear GCN as the surrogate model.

**Main results.** We attack individual fairness using FATE via both edge flipping (FATE-flip in Table 8 and edge addition (FATE-add in Table 8), whereas edges are only added for all other baseline methods. From Table 8, we can see that: (1) for *Pokec-n* and *Pokec-z*, FATE-flip and FATE-add are effective: they are the only methods that could consistently attack individual fairness across different perturbation rates; FATE-flip and FATE-add are deceptive by achieving comparable or higher micro F1 scores compared with the micro F1 score on the benign graph (when perturbation rate is 0.00). (2) For *Bail*, almost all methods fail the fairness attacks. A possible reason is that the adjacency matrix $\mathbf{A}$ of *Bail* is essentially a similarity graph, which causes pairwise node similarity matrix $\mathbf{S}$ being close to the adjacency matrix $\mathbf{A}$. Even though FATE (and other baseline methods) add adversarial edges to attack individual fairness, regularizing the individual bias defined by $\mathbf{S}$ (a) not only helps to ensure individual fairness (b) but also provide useful supervision signal in learning a representative node representation due to the closeness between $\mathbf{S}$ and $\mathbf{A}$. (3) Compared with the results in Table 2 where GCN is the victim model, InFoRM-GNN is more robust against fairness attacks against individual fairness due to smaller individual bias in Table 8.

**Effect of the perturbation rate.** From Table 8, we can see that FATE can always achieve comparable or even better micro F1 scores across different perturbation rates. In the meanwhile, the correlation between the perturbation rate and the individual bias is relatively weak. One possible reason is that the individual bias is computed using the pairwise node similarity matrix, which is not impacted by poisoning the adjacency matrix. Though poisoning the adjacency matrix could affect the learning results, the goal of achieving deceptive fairness attacks (i.e., the lower-level optimization problem in FATE) may not cause the learning results obtained by training on the benign graph to deviate much from the learning results obtained by training on the poisoned graph. Consequently, a higher perturbation rate may have less impact on the computation of individual bias.

Table 8: Effectiveness of attacking individual fairness on InFoRM-GNN. FATE poisons the graph via both edge flipping (FATE-flip) and edge addition (FATE-add) while all other baselines poison the graph via edge addition. Higher is better (↑) for micro F1 score (Micro F1) and InFoRM bias (Bias). Bold font indicates the success of fairness attack (i.e., bias is increased after attack) with the highest micro F1 score. Underlined cell indicates the failure of fairness attack (i.e., $\Delta_{\text{SP}}$ is decreased after attack).

| Dataset | Ptb. | Random | | DICE | | FA-GNN | | FATE-flip | | FATE-add | |
|---|---|---|---|---|---|---|---|---|---|---|---|
| | | Micro F1 (↑) | Bias (↑) | Micro F1 (↑) | Bias (↑) | Micro F1 (↑) | Bias (↑) | Micro F1 (↑) | Bias (↑) | Micro F1 (↑) | Bias (↑) |
| Pokec-n | 0.00 | $68.0 \pm 0.4$ | $0.5 \pm 0.1$ | $68.0 \pm 0.4$ | $0.5 \pm 0.1$ | $68.0 \pm 0.4$ | $0.5 \pm 0.1$ | $68.0 \pm 0.4$ | $0.5 \pm 0.1$ | $68.0 \pm 0.4$ | $0.5 \pm 0.1$ |
| | 0.05 | $67.3 \pm 0.5$ | $0.5 \pm 0.0$ | $67.0 \pm 0.2$ | $0.5 \pm 0.0$ | $68.3 \pm 0.2$ | $0.5 \pm 0.0$ | $\mathbf{68.4 \pm 0.4}$ | $\mathbf{0.6 \pm 0.1}$ | $68.3 \pm 0.4$ | $0.5 \pm 0.1$ |
| | 0.10 | $67.0 \pm 0.2$ | $0.5 \pm 0.1$ | $65.5 \pm 0.6$ | $0.5 \pm 0.1$ | $67.2 \pm 0.2$ | $0.4 \pm 0.0$ | $68.3 \pm 0.6$ | $0.5 \pm 0.1$ | $\mathbf{68.4 \pm 0.5}$ | $\mathbf{0.6 \pm 0.1}$ |
| | 0.15 | $66.7 \pm 0.5$ | $0.5 \pm 0.1$ | $63.8 \pm 0.3$ | $\underline{0.4 \pm 0.0}$ | $66.1 \pm 0.2$ | $\underline{0.4 \pm 0.0}$ | $\mathbf{68.3 \pm 0.6}$ | $\mathbf{0.6 \pm 0.1}$ | $68.1 \pm 0.7$ | $0.6 \pm 0.1$ |
| | 0.20 | $66.9 \pm 0.3$ | $\underline{0.4 \pm 0.1}$ | $63.7 \pm 0.2$ | $\underline{0.3 \pm 0.0}$ | $66.5 \pm 0.2$ | $\underline{0.4 \pm 0.0}$ | $67.9 \pm 0.8$ | $0.5 \pm 0.1$ | $\mathbf{68.1 \pm 0.7}$ | $\mathbf{0.6 \pm 0.1}$ |
| | 0.25 | $66.6 \pm 0.5$ | $0.5 \pm 0.0$ | $62.2 \pm 0.5$ | $\underline{0.2 \pm 0.1}$ | $65.1 \pm 0.2$ | $\underline{0.4 \pm 0.0}$ | $\mathbf{68.7 \pm 0.3}$ | $\mathbf{0.6 \pm 0.0}$ | $68.5 \pm 0.8$ | $0.6 \pm 0.1$ |
| Pokec-z | 0.00 | $68.4 \pm 0.5$ | $0.5 \pm 0.0$ | $68.4 \pm 0.5$ | $0.5 \pm 0.0$ | $68.4 \pm 0.5$ | $0.5 \pm 0.0$ | $68.4 \pm 0.5$ | $0.5 \pm 0.0$ | $68.4 \pm 0.5$ | $0.5 \pm 0.0$ |
| | 0.05 | $68.9 \pm 0.2$ | $0.6 \pm 0.1$ | $67.0 \pm 0.6$ | $0.6 \pm 0.1$ | $68.1 \pm 0.7$ | $0.5 \pm 0.1$ | $68.7 \pm 0.7$ | $0.7 \pm 0.1$ | $\mathbf{68.9 \pm 0.5}$ | $\mathbf{0.6 \pm 0.0}$ |
| | 0.10 | $67.9 \pm 0.2$ | $0.6 \pm 0.1$ | $66.3 \pm 0.4$ | $0.5 \pm 0.1$ | $68.0 \pm 0.6$ | $0.5 \pm 0.0$ | $\mathbf{68.9 \pm 0.6}$ | $\mathbf{0.6 \pm 0.0}$ | $68.8 \pm 0.6$ | $0.6 \pm 0.0$ |
| | 0.15 | $67.6 \pm 0.3$ | $0.6 \pm 0.1$ | $65.3 \pm 0.4$ | $\underline{0.4 \pm 0.1}$ | $66.8 \pm 0.3$ | $0.5 \pm 0.1$ | $\mathbf{69.1 \pm 0.5}$ | $\mathbf{0.6 \pm 0.0}$ | $69.0 \pm 0.7$ | $0.6 \pm 0.1$ |
| | 0.20 | $67.7 \pm 0.5$ | $0.6 \pm 0.1$ | $63.9 \pm 0.6$ | $\underline{0.3 \pm 0.0}$ | $66.4 \pm 0.6$ | $\underline{0.4 \pm 0.1}$ | $69.1 \pm 0.2$ | $0.6 \pm 0.0$ | $\mathbf{69.3 \pm 0.3}$ | $\mathbf{0.6 \pm 0.0}$ |
| | 0.25 | $66.8 \pm 0.4$ | $0.5 \pm 0.1$ | $64.5 \pm 0.3$ | $\underline{0.2 \pm 0.0}$ | $65.3 \pm 0.4$ | $\underline{0.4 \pm 0.0}$ | $68.9 \pm 0.7$ | $0.6 \pm 0.0$ | $\mathbf{69.4 \pm 0.4}$ | $\mathbf{0.6 \pm 0.0}$ |
| Bail | 0.00 | $92.8 \pm 0.1$ | $1.7 \pm 0.1$ | $92.8 \pm 0.1$ | $1.7 \pm 0.1$ | $92.8 \pm 0.1$ | $1.7 \pm 0.1$ | $92.8 \pm 0.1$ | $1.7 \pm 0.1$ | $92.8 \pm 0.1$ | $1.7 \pm 0.1$ |
| | 0.05 | $91.9 \pm 0.1$ | $\underline{0.4 \pm 0.0}$ | $91.5 \pm 0.1$ | $\underline{1.6 \pm 0.1}$ | $91.3 \pm 0.1$ | $\underline{1.5 \pm 0.1}$ | $92.8 \pm 0.3$ | $\underline{1.7 \pm 0.1}$ | $92.7 \pm 0.1$ | $\underline{1.6 \pm 0.1}$ |
| | 0.10 | $91.7 \pm 0.1$ | $\underline{0.3 \pm 0.0}$ | $90.4 \pm 0.1$ | $\underline{1.4 \pm 0.1}$ | $90.4 \pm 0.2$ | $\underline{1.5 \pm 0.1}$ | $92.8 \pm 0.1$ | $\underline{1.6 \pm 0.0}$ | $92.8 \pm 0.1$ | $\underline{1.6 \pm 0.0}$ |
| | 0.15 | $91.5 \pm 0.1$ | $\underline{0.3 \pm 0.0}$ | $89.7 \pm 0.1$ | $\underline{1.4 \pm 0.0}$ | $90.0 \pm 0.1$ | $\underline{1.7 \pm 0.1}$ | $92.8 \pm 0.0$ | $\underline{1.6 \pm 0.1}$ | $92.8 \pm 0.1$ | $\underline{1.6 \pm 0.0}$ |
| | 0.20 | $91.5 \pm 0.1$ | $\underline{0.3 \pm 0.0}$ | $88.6 \pm 0.2$ | $\underline{1.3 \pm 0.1}$ | $\mathbf{89.1 \pm 0.1}$ | $\mathbf{1.7 \pm 0.1}$ | $92.8 \pm 0.1$ | $\underline{1.6 \pm 0.0}$ | $92.7 \pm 0.1$ | $\underline{1.5 \pm 0.1}$ |
| | 0.25 | $91.1 \pm 0.2$ | $\underline{0.3 \pm 0.0}$ | $87.6 \pm 0.2$ | $\underline{1.3 \pm 0.1}$ | $\mathbf{88.9 \pm 0.1}$ | $\mathbf{1.8 \pm 0.1}$ | $92.6 \pm 0.1$ | $\underline{1.6 \pm 0.1}$ | $92.7 \pm 0.0$ | $\underline{1.6 \pm 0.1}$ |

**B – Performance evaluation under different utility metrics.** Similar to Appendix D, we provide additional results on evaluating the utility of FATE in attacking individual fairness with macro F1 score and AUC score. From Tables 9 and 10, we can draw a conclusion that FATE can achieve comparable or even better macro F1 scores and AUC scores for both GCN and InFoRM-GNN across different perturbation rates. It further proves the ability of FATE on deceptive fairness attacks in the task of semi-supervised node classification.

Table 9: Macro F1 score and AUC score of attacking individual fairness on GCN. FATE poisons the graph via both edge flipping (FATE-flip) and edge addition (FATE-add) while all other baselines poison the graph via edge addition. Higher is better (↑) for macro F1 score (Macro F1) and AUC score (AUC). Bold font indicates the highest macro F1 score or AUC score.

| Dataset | Ptb. | Random | | DICE | | FA-GNN | | FATE-flip | | FATE-add | |
|---|---|---|---|---|---|---|---|---|---|---|---|
| | | Macro F1 (↑) | AUC (↑) | Macro F1 (↑) | AUC (↑) | Macro F1 (↑) | AUC (↑) | Macro F1 (↑) | AUC (↑) | Macro F1 (↑) | AUC (↑) |
| Pokec-n | 0.00 | 65.3 ± 0.3 | 69.9 ± 0.5 | 65.3 ± 0.4 | 70.5 ± 0.8 | 65.3 ± 0.4 | 69.9 ± 0.5 | 65.3 ± 0.3 | 69.9 ± 0.5 | 65.3 ± 0.3 | 69.9 ± 0.5 |
| | 0.05 | 65.2 ± 0.3 | 70.1 ± 0.2 | 64.4 ± 0.4 | 69.5 ± 0.2 | 65.6 ± 0.6 | **71.1 ± 0.2** | **65.7 ± 0.4** | 70.1 ± 0.6 | 65.5 ± 0.3 | 70.2 ± 0.8 |
| | 0.10 | 65.2 ± 0.3 | 69.6 ± 0.5 | 62.5 ± 0.3 | 66.9 ± 0.2 | 65.4 ± 0.6 | 70.2 ± 0.3 | 65.5 ± 0.3 | 70.2 ± 0.7 | **65.8 ± 0.5** | **70.7 ± 0.6** |
| | 0.15 | 65.4 ± 0.2 | 69.4 ± 0.3 | 60.9 ± 0.5 | 64.8 ± 0.4 | 64.6 ± 0.2 | 69.4 ± 0.1 | **65.6 ± 0.4** | **70.0 ± 0.5** | 65.4 ± 0.1 | 69.8 ± 0.7 |
| | 0.20 | 64.9 ± 0.2 | 69.6 ± 0.3 | 61.3 ± 0.2 | 65.5 ± 0.2 | 63.7 ± 0.5 | 69.0 ± 0.1 | 65.2 ± 0.3 | 69.7 ± 0.6 | **65.6 ± 0.6** | **70.2 ± 0.7** |
| | 0.25 | 64.7 ± 0.1 | 69.4 ± 0.2 | 60.0 ± 0.3 | 63.5 ± 0.4 | 63.3 ± 0.5 | 68.4 ± 0.3 | 65.4 ± 0.6 | 69.7 ± 0.7 | **65.6 ± 0.8** | **69.8 ± 0.8** |
| Pokec-z | 0.00 | 68.2 ± 0.4 | 75.1 ± 0.3 | 68.2 ± 0.4 | 75.1 ± 0.3 | 68.2 ± 0.4 | 75.1 ± 0.3 | 68.2 ± 0.4 | 75.1 ± 0.3 | 68.2 ± 0.4 | 75.1 ± 0.3 |
| | 0.05 | **68.7 ± 0.4** | 75.0 ± 0.4 | 67.0 ± 0.5 | 73.8 ± 0.5 | 68.0 ± 0.4 | 75.1 ± 0.5 | 68.5 ± 0.5 | **75.4 ± 0.2** | 68.5 ± 0.3 | 75.2 ± 0.4 |
| | 0.10 | 68.5 ± 0.1 | 75.1 ± 0.5 | 66.0 ± 0.5 | 72.5 ± 0.3 | 67.9 ± 0.6 | 74.4 ± 0.5 | 68.8 ± 0.5 | 75.5 ± 0.3 | **68.8 ± 0.4** | **75.6 ± 0.2** |
| | 0.15 | 67.5 ± 0.4 | 74.4 ± 0.3 | 65.1 ± 0.4 | 71.0 ± 0.5 | 66.8 ± 0.4 | 72.6 ± 0.2 | 68.4 ± 0.5 | 75.5 ± 0.3 | **68.8 ± 0.7** | **75.6 ± 0.3** |
| | 0.20 | 67.5 ± 0.4 | 74.7 ± 0.4 | 63.9 ± 0.3 | 69.4 ± 0.4 | 66.1 ± 0.1 | 71.8 ± 0.2 | 68.7 ± 0.5 | 75.5 ± 0.3 | **69.0 ± 0.4** | **75.6 ± 0.3** |
| | 0.25 | 67.2 ± 0.3 | 74.1 ± 0.3 | 63.5 ± 0.4 | 68.5 ± 0.3 | 64.8 ± 0.4 | 70.5 ± 0.4 | 68.9 ± 0.3 | 75.6 ± 0.2 | **69.1 ± 0.3** | **75.7 ± 0.3** |
| Bail | 0.00 | 92.3 ± 0.2 | 97.4 ± 0.1 | 92.3 ± 0.2 | 97.4 ± 0.1 | 92.3 ± 0.2 | 97.4 ± 0.1 | 92.3 ± 0.2 | 97.4 ± 0.1 | 92.3 ± 0.2 | 97.4 ± 0.1 |
| | 0.05 | 91.2 ± 0.3 | 94.8 ± 0.2 | 90.9 ± 0.1 | 94.0 ± 0.2 | 90.3 ± 0.2 | 94.1 ± 0.2 | **92.3 ± 0.4** | **97.3 ± 0.1** | 92.1 ± 0.3 | **97.3 ± 0.1** |
| | 0.10 | 90.6 ± 0.1 | 94.2 ± 0.3 | 89.0 ± 0.1 | 91.9 ± 0.2 | 89.1 ± 0.1 | 92.9 ± 0.3 | **92.3 ± 0.1** | **97.3 ± 0.4** | 92.2 ± 0.2 | **97.3 ± 0.1** |
| | 0.15 | 90.3 ± 0.1 | 94.1 ± 0.2 | 88.0 ± 0.0 | 90.9 ± 0.2 | 88.6 ± 0.2 | 92.4 ± 0.3 | **92.4 ± 0.1** | **97.3 ± 0.0** | 92.3 ± 0.2 | **97.3 ± 0.1** |
| | 0.20 | 90.2 ± 0.0 | 93.9 ± 0.1 | 87.1 ± 0.2 | 90.4 ± 0.2 | 87.9 ± 0.2 | 91.8 ± 0.2 | **92.4 ± 0.1** | **97.3 ± 0.0** | **92.4 ± 0.2** | **97.3 ± 0.1** |
| | 0.25 | 90.9 ± 0.1 | 93.5 ± 0.2 | 85.9 ± 0.3 | 89.6 ± 0.3 | 87.6 ± 0.1 | 91.6 ± 0.2 | **92.2 ± 0.2** | 97.2 ± 0.1 | **92.2 ± 0.2** | **97.3 ± 0.1** |

Table 10: Macro F1 score and AUC score of attacking individual fairness on InFoRM-GNN. FATE poisons the graph via both edge flipping (FATE-flip) and edge addition (FATE-add) while all other baselines poison the graph via edge addition. Higher is better (↑) for macro F1 score (Macro F1) and AUC score (AUC). Bold font indicates the highest macro F1 score or AUC score.

| Dataset | Ptb. | Random | | DICE | | FA-GNN | | FATE-flip | | FATE-add | |
|---|---|---|---|---|---|---|---|---|---|---|---|
| | | Macro F1 (↑) | AUC (↑) | Macro F1 (↑) | AUC (↑) | Macro F1 (↑) | AUC (↑) | Macro F1 (↑) | AUC (↑) | Macro F1 (↑) | AUC (↑) |
| Pokec-n | 0.00 | 65.4 ± 0.4 | 70.5 ± 0.8 | 65.4 ± 0.4 | 70.5 ± 0.8 | 65.4 ± 0.4 | 70.5 ± 0.8 | 65.4 ± 0.4 | 70.5 ± 0.8 | 65.4 ± 0.4 | 70.5 ± 0.8 |
| | 0.05 | 65.1 ± 0.3 | 69.9 ± 0.2 | 64.5 ± 0.4 | 69.3 ± 0.2 | 65.6 ± 0.3 | **70.8 ± 0.1** | **65.9 ± 0.4** | 70.7 ± 0.8 | 65.8 ± 0.5 | 70.5 ± 0.9 |
| | 0.10 | 64.9 ± 0.2 | 69.6 ± 0.5 | 62.7 ± 0.4 | 67.0 ± 0.2 | 64.8 ± 0.4 | 69.8 ± 0.3 | 65.8 ± 0.5 | 70.3 ± 1.0 | **66.0 ± 0.4** | **70.9 ± 1.1** |
| | 0.15 | 64.8 ± 0.4 | 69.6 ± 0.4 | 60.7 ± 0.2 | 64.8 ± 0.2 | 64.4 ± 0.1 | 69.2 ± 0.3 | 65.7 ± 0.6 | **70.3 ± 0.7** | 65.8 ± 0.4 | **70.3 ± 0.9** |
| | 0.20 | 65.1 ± 0.2 | 69.5 ± 0.3 | 61.0 ± 0.1 | 65.4 ± 0.3 | 63.4 ± 0.4 | 69.0 ± 0.2 | 65.5 ± 0.8 | 70.2 ± 0.9 | 65.6 ± 0.6 | **70.5 ± 0.7** |
| | 0.25 | 64.6 ± 0.3 | 69.6 ± 0.2 | 59.8 ± 0.3 | 63.4 ± 0.2 | 63.6 ± 0.3 | 68.6 ± 0.2 | **66.0 ± 0.5** | **70.8 ± 0.3** | 65.9 ± 0.5 | 70.4 ± 0.8 |
| Pokec-z | 0.00 | 68.3 ± 0.4 | 75.2 ± 0.2 | 68.3 ± 0.4 | 75.2 ± 0.2 | 68.3 ± 0.4 | 75.2 ± 0.2 | 68.3 ± 0.4 | 75.2 ± 0.2 | 68.3 ± 0.4 | 75.2 ± 0.2 |
| | 0.05 | 68.6 ± 0.2 | 75.1 ± 0.3 | 66.9 ± 0.5 | 73.4 ± 0.4 | 67.8 ± 0.6 | 75.0 ± 0.3 | 68.6 ± 0.7 | 75.1 ± 0.6 | **68.7 ± 0.4** | **75.4 ± 0.3** |
| | 0.10 | 67.6 ± 0.2 | 74.3 ± 0.4 | 66.1 ± 0.4 | 72.1 ± 0.8 | 67.7 ± 0.6 | 73.9 ± 0.6 | 68.6 ± 0.6 | **75.6 ± 0.3** | 68.6 ± 0.6 | 75.5 ± 0.3 |
| | 0.15 | 67.2 ± 0.3 | 74.1 ± 0.4 | 64.9 ± 0.5 | 71.2 ± 0.3 | 66.7 ± 0.3 | 72.3 ± 0.1 | **68.9 ± 0.4** | 75.4 ± 0.4 | **68.9 ± 0.6** | 75.4 ± 0.4 |
| | 0.20 | 67.3 ± 0.6 | 74.4 ± 0.4 | 63.9 ± 0.6 | 69.4 ± 0.5 | 66.0 ± 0.5 | 71.7 ± 0.2 | 69.0 ± 0.2 | **75.5 ± 0.2** | **69.2 ± 0.4** | 75.4 ± 0.4 |
| | 0.25 | 66.3 ± 0.4 | 73.9 ± 0.4 | 63.9 ± 0.6 | 68.9 ± 0.4 | 65.0 ± 0.5 | 70.8 ± 0.2 | 68.8 ± 0.7 | 75.6 ± 0.3 | **69.3 ± 0.4** | **75.8 ± 0.1** |
| Bail | 0.00 | 91.9 ± 0.1 | 97.2 ± 0.0 | 91.9 ± 0.1 | 97.2 ± 0.0 | 91.9 ± 0.1 | 97.2 ± 0.0 | 91.9 ± 0.1 | 97.2 ± 0.0 | 91.9 ± 0.1 | 97.2 ± 0.0 |
| | 0.05 | 91.0 ± 0.1 | 94.2 ± 0.2 | 90.5 ± 0.1 | 93.9 ± 0.1 | 90.4 ± 0.1 | 94.2 ± 0.1 | **92.0 ± 0.0** | **97.1 ± 0.1** | 91.9 ± 0.2 | 97.0 ± 0.2 |
| | 0.10 | 90.7 ± 0.2 | 93.9 ± 0.3 | 89.3 ± 0.1 | 92.4 ± 0.3 | 89.4 ± 0.2 | 93.3 ± 0.1 | **92.0 ± 0.1** | **97.0 ± 0.0** | 91.9 ± 0.1 | **97.0 ± 0.0** |
| | 0.15 | 90.5 ± 0.1 | 94.1 ± 0.2 | 88.4 ± 0.1 | 91.4 ± 0.3 | 88.8 ± 0.2 | 92.4 ± 0.1 | **92.0 ± 0.1** | **97.0 ± 0.0** | 91.9 ± 0.2 | **97.0 ± 0.1** |
| | 0.20 | 90.5 ± 0.2 | 93.7 ± 0.2 | 87.3 ± 0.2 | 90.5 ± 0.3 | 87.8 ± 0.1 | 91.8 ± 0.1 | **92.0 ± 0.1** | **96.9 ± 0.0** | 91.9 ± 0.1 | 96.8 ± 0.1 |
| | 0.25 | 90.1 ± 0.2 | 93.4 ± 0.3 | 85.9 ± 0.2 | 89.1 ± 0.5 | 87.4 ± 0.1 | 91.4 ± 0.1 | 91.8 ± 0.1 | 96.8 ± 0.1 | **91.9 ± 0.1** | **96.9 ± 0.0** |

# F   Tranferability of Fairness Attacks by FATE

For the evaluation results shown in Sections 6.1 and 6.2 as well as Appendices D and E, both the surrogate model (linear GCN) and the victim models (i.e., GCN, FairGNN, InFoRM-GNN) are convolutional aggregation-based graph neural networks. In this section, we aim to test the transferability of FATE by generating poisoned graphs on the convolutional aggregation-based surrogate model (i.e., linear GCN) and testing on graph attention network (GAT), which is a non-convolutional aggregation-based graph neural network.

More specifically, we train a graph attention network (GAT) with $8$ attention heads for $400$ epochs. The hidden dimension, learning rate, weight decay and dropout rate of GAT are set to $64$, $1e-3$, $1e-5$ and $0.5$, respectively.

The results on attacking statistical parity and individual fairness with GAT as the victim model are shown in Table 11. Even though the surrogate model used by the attacker is a convolutional aggregation-based linear GCN, from the table, it is clear that FATE can consistently succeed in (1) effective fairness attack by increasing $\Delta_{SP}$ and the individual bias (Bias) and (2) deceptive attack by offering comparable or even better micro F1 score (Micro F1) when the victim model is not a convolutional aggregation-based model. Thus, it shows that the adversarial edges flipped/added by FATE is able to transfer to graph neural networks with different type of aggregation function.

Table 11: Transferability of attacking statical parity and individual fairness with FATE on GAT. FATE poisons the graph via both edge flipping (FATE-flip) and edge addition (FATE-add). Higher is better (↑) for micro F1 score (Micro F1) $\Delta_{SP}$ and InFoRM bias (Bias).

| Attacking Statistical Parity | | | | | | | |
|---|---|---|---|---|---|---|---|
| **Dataset** | **Ptb.** | **Pokec-n** | | **Pokec-z** | | **Bail** | |
| | | **Micro F1 (↑)** | **Bias (↑)** | **Micro F1 (↑)** | **$\Delta_{SP}$ (↑)** | **Micro F1 (↑)** | **$\Delta_{SP}$ (↑)** |
| **FATE-flip** | 0.00 | $63.8 \pm 5.3$ | $4.0 \pm 3.2$ | $68.2 \pm 0.5$ | $8.6 \pm 1.1$ | $89.7 \pm 4.2$ | $7.5 \pm 0.6$ |
| | 0.05 | $63.9 \pm 5.5$ | $6.4 \pm 5.1$ | $68.3 \pm 0.4$ | $10.5 \pm 1.3$ | $90.1 \pm 3.8$ | $8.1 \pm 0.6$ |
| | 0.10 | $63.6 \pm 5.3$ | $7.9 \pm 6.7$ | $67.8 \pm 0.4$ | $11.2 \pm 1.7$ | $90.3 \pm 3.2$ | $8.5 \pm 0.6$ |
| | 0.15 | $63.7 \pm 5.3$ | $7.5 \pm 6.1$ | $68.2 \pm 0.6$ | $11.2 \pm 1.5$ | $90.2 \pm 2.7$ | $8.8 \pm 0.3$ |
| | 0.20 | $64.1 \pm 5.6$ | $7.7 \pm 6.3$ | $67.8 \pm 0.6$ | $11.1 \pm 0.9$ | $90.0 \pm 2.7$ | $8.7 \pm 0.6$ |
| | 0.25 | $63.6 \pm 5.2$ | $8.5 \pm 7.0$ | $68.0 \pm 0.4$ | $11.5 \pm 1.2$ | $89.9 \pm 3.0$ | $8.8 \pm 0.5$ |
| **FATE-add** | 0.00 | $63.8 \pm 5.3$ | $4.0 \pm 3.2$ | $68.2 \pm 0.5$ | $8.6 \pm 1.1$ | $89.7 \pm 4.2$ | $7.5 \pm 0.6$ |
| | 0.05 | $63.9 \pm 5.5$ | $6.4 \pm 5.1$ | $68.3 \pm 0.4$ | $10.5 \pm 1.3$ | $90.2 \pm 3.7$ | $8.1 \pm 0.7$ |
| | 0.10 | $63.6 \pm 5.3$ | $7.9 \pm 6.7$ | $67.8 \pm 0.4$ | $11.2 \pm 1.7$ | $90.3 \pm 3.2$ | $8.5 \pm 0.6$ |
| | 0.15 | $63.7 \pm 5.3$ | $7.5 \pm 6.1$ | $68.2 \pm 0.6$ | $11.2 \pm 1.5$ | $90.3 \pm 2.6$ | $8.8 \pm 0.3$ |
| | 0.20 | $64.1 \pm 5.6$ | $7.7 \pm 6.3$ | $67.8 \pm 0.6$ | $11.1 \pm 0.9$ | $90.1 \pm 2.6$ | $8.8 \pm 0.5$ |
| | 0.25 | $63.6 \pm 5.2$ | $8.5 \pm 7.0$ | $68.0 \pm 0.4$ | $11.5 \pm 1.2$ | $89.9 \pm 2.9$ | $8.8 \pm 0.5$ |
| **Attacking Individual Fairness** | | | | | | | |
| **Dataset** | **Ptb.** | **Pokec-n** | | **Pokec-z** | | **Bail** | |
| | | **Micro F1 (↑)** | **Bias (↑)** | **Micro F1 (↑)** | **Bias (↑)** | **Micro F1 (↑)** | **Bias (↑)** |
| **FATE-flip** | 0.00 | $63.8 \pm 5.3$ | $0.4 \pm 0.2$ | $68.2 \pm 0.5$ | $0.5 \pm 0.1$ | $89.7 \pm 4.2$ | $2.5 \pm 1.2$ |
| | 0.05 | $63.6 \pm 5.3$ | $0.5 \pm 0.2$ | $68.2 \pm 0.8$ | $0.6 \pm 0.1$ | $90.0 \pm 4.2$ | $2.7 \pm 1.1$ |
| | 0.10 | $63.7 \pm 5.3$ | $0.5 \pm 0.2$ | $67.8 \pm 0.5$ | $0.6 \pm 0.1$ | $90.0 \pm 4.0$ | $2.8 \pm 1.3$ |
| | 0.15 | $63.7 \pm 5.4$ | $0.5 \pm 0.2$ | $68.2 \pm 0.5$ | $0.6 \pm 0.2$ | $90.2 \pm 3.6$ | $2.8 \pm 1.4$ |
| | 0.20 | $63.5 \pm 5.1$ | $0.5 \pm 0.2$ | $68.5 \pm 0.5$ | $0.6 \pm 0.2$ | $90.2 \pm 3.4$ | $2.8 \pm 1.2$ |
| | 0.25 | $63.5 \pm 5.1$ | $0.5 \pm 0.2$ | $68.0 \pm 0.6$ | $0.6 \pm 0.1$ | $90.2 \pm 3.1$ | $2.7 \pm 1.2$ |
| **FATE-add** | 0.00 | $63.8 \pm 5.3$ | $0.4 \pm 0.2$ | $68.2 \pm 0.5$ | $0.5 \pm 0.1$ | $89.7 \pm 4.2$ | $2.5 \pm 1.2$ |
| | 0.05 | $63.9 \pm 5.4$ | $0.5 \pm 0.2$ | $68.2 \pm 0.7$ | $0.6 \pm 0.1$ | $90.0 \pm 4.6$ | $2.7 \pm 1.4$ |
| | 0.10 | $63.8 \pm 5.4$ | $0.5 \pm 0.2$ | $68.2 \pm 0.5$ | $0.6 \pm 0.2$ | $90.1 \pm 4.0$ | $2.8 \pm 1.2$ |
| | 0.15 | $63.8 \pm 5.4$ | $0.5 \pm 0.2$ | $68.3 \pm 0.2$ | $0.6 \pm 0.2$ | $90.1 \pm 3.9$ | $2.8 \pm 1.2$ |
| | 0.20 | $63.7 \pm 5.3$ | $0.5 \pm 0.2$ | $68.4 \pm 0.3$ | $0.6 \pm 0.1$ | $90.3 \pm 3.2$ | $2.8 \pm 1.3$ |
| | 0.25 | $63.7 \pm 5.3$ | $0.5 \pm 0.2$ | $68.4 \pm 0.3$ | $0.6 \pm 0.1$ | $90.2 \pm 3.1$ | $2.8 \pm 1.2$ |

## G Further Discussions about FATE

**A – Relationship between fairness attacks and the impossibility theorem of fairness.** The impossibility theorems show that some fairness definitions may not be satisfied at the same time.[6] However, this may not always be regarded as fairness attacks. To our best knowledge, the impossibility theorems prove that two fairness definitions (e.g., statistical parity and predictive parity) cannot be fully satisfied at the same time, i.e., biases for two fairness definitions are both zero). However, there is no formal theoretical guarantees that ensuring one fairness definition will *always* amplify the bias of another fairness definition. Such formal guarantees might be nontrivial and beyond the scope of our paper. As we pointed out in the abstract, the main goal of this paper is to provide insights into the adversarial robustness of fair graph learning and can shed light for designing robust and fair graph learning in future studies.

**B – Relationship between FATE and Metattack.** FATE bears subtle differences with Metattack [50], which also utilize meta learning for adversarial attacks. Note that Metattack aims to degrade the utility of a graph neural network by maximizing the task-specific utility loss (e.g., cross entropy for node classification) in the upper-level optimization problem. Different from Metattack, FATE aims to attack the fairness instead of utility by setting the upper-level optimization problem as maximizing a bias function rather than a task-specific utility loss.

---

[6] https://machinesgonewrong.com/fairness/