# OpenReview forum: "FATE: Fairness Attacks on Graph Learning"
_NeurIPS.cc/2023/Conference — Submitted to NeurIPS 2023_

### Official Review · Reviewer_NH2N · 2023-07-04

**Soundness:** 3 good
**Presentation:** 3 good
**Contribution:** 2 fair
**Rating:** 5
**Confidence:** 4

**Summary:**

This paper proposes to attack different fairness definitions for a variety of graph learning models. The task is formulated as a bi-level optimization problem, which is solved in a meta learning manner. The major advantages of the proposed framework include 1) it is feasible to any fairness notion and graph learning model, 2) it can support continuous and discrete perturbation on the graph topology. Experiments demonstrate the attack efficacy and classification utility of the proposed method.

**Strengths:**

1. The proposed framework provides a good coverage for multiple types of (differentiable) fairness notions (e.g., statistical parity, individual fairness), and graph learning models (e.g., non-parametric, parametric);
2. Extensive experiments are conducted, considering possible fairness defenses (FairGNN and InFoRM-GNN) and transferability.

**Weaknesses:**

1. The clarity and rationale of methodology can be improved: there are inconsistent definitions (of the budget constraint), and the realization of budget constraint in the optimization procedure is not validated. See detailed questions;
2. The IID assumption for kernel density estimation may not hold in graph data;
3. Important baseline should be compared (a modified version of DICE based on the sensitive group); and more related works should be discussed.

**Questions:**

I appreciate the breath of this work which makes it of good potential. However, the authors are encouraged to address the following questions to improve the technical depth of this work.

**Methodology**

1. The realization of the budget constraint needs more justification. In line 180 for continuous cases, only limiting the learning rate cannot guarantee the final perturbation is under budget B (after several steps of gradient descent). The authors should either provide citation/reason to justify the validity of this design, or refer to the better justified technique – projected gradient descent. In Line 181-188 for discrete case, the approximation is also questionable: other two cases are directly discarded (i.e., $\Delta_A b[i, j]>0, (1-2A)[i,j]=-1$, and $\Delta_A b[i, j]<0, (1-2A)[i,j]=1$), which could result in large rounding error. The authors are recommended to check and discuss recent discrete attack techniques for graphs [1].

2. The kernel density estimation in Eq. (9) assumes that samples are IID, which however does not hold for graph data, as nodes are connected and non-IID. Can the authors provide further justification?

**Experiment**

3. One important baseline should be considered and compared: modifying DICE to consider communities based on nodes’ sensitive groups rather than their classification labels, so this heuristic baseline will add edges between nodes which belong to the same sensitive group and/or remove edges between nodes which are in different sensitive groups.

**Related work**

4. This work provides additional experiments on possible fairness defenses (i.e., FairGNN and InFoRM-GNN), which is highly appreciated. Beyond such works that are specified to a certain fairness definition, the authors are encouraged to discuss and/or test on the recent debiasing work [2], which does not rely on specific fairness definitions.

**Minors**

5. The definition of perturbation budget is not consistent in Line 98 and Line 121. Meanwhile, the “norm” in Line 98 is not in a typical math form: what is $\|A, \tilde{A}\|_{1, 1}$?

6. The budget defined in Line 121, $\|A- \tilde{A}\|_{1, 1}$, is mathematically the maximum absolute column sum of the matrix difference, which seems to be inconsistent with the attacker’s capability to “perturb up to B edges in the graph”. Meanwhile, based on Eq. (8) which limits the number of total edge changes, the budget definition should be corrected as zero norm, right?

7. The bias function $b()$ is not consistent in Line 94 and Line 99. The hyperparameter $\theta$ should not be an input to the function, right?

8. In Table 2, $\Delta_{SP}$ -> InFoRM bias.

**References**

[1] Lin, Lu, et al. "Graph structural attack by perturbing spectral distance." Proceedings of the 28th ACM SIGKDD Conference on Knowledge Discovery and Data Mining. 2022.

[2] Wang, Nan, et al. "Unbiased graph embedding with biased graph observations." Proceedings of the ACM Web Conference 2022. 2022.

**Limitations:**

The authors provide reasonable discussion about the limitations of this work (i.e., other fairness notions, space efficiency).

---

> ### Author Rebuttal · Authors · 2023-08-10
>
> We appreciate your constructive comments and valuable feedback for further improving our work. We summarize the main concerns and the point-to-point responses as follows.
>
> **Q1. Justification of learning rate.**
>
> Thank you for your comments. We would like to clarify that Eq. (6) discusses continuous attack in one step. When we consider multiple steps of attack, the update rule would be $\mathbf{A} \leftarrow \mathbf{A} - \eta_i \nabla_{\mathbf{A} b}$ where $\mathbf{A}$ is the input graph, $\eta_i$ is the learning rate in $i$-th step that should satisfy $\eta_i \leq \frac{\delta_i}{\left\|\nabla_{\mathbf{A}} b\right\|}$, and $\delta_i$ is the budget in $i$-th step. We will clarify it in the revised version.
>
> **Q2. Two discarded cases in discretized attack**
>
> Thank you for your comments. In our two discussed cases, when $\Delta_{\mathbf{A}}b$ is positive, it indicates that adding an edge will help increase the bias. Since $\left(\mathbf{I}-2\mathbf{A}\right)\left[i,j\right] > 0$ means adding an edge, positive $\Delta_{\mathbf{A}}b$ and $\left(\mathbf{I}-2\mathbf{A}\right)\left[i,j\right]>0$ indicates a strong preference in adding edge $\left(i,j\right)$ to increase the bias. However, if $\left(\mathbf{I}-2\mathbf{A}\right)\left[i,j\right]<0$, it suggests removing edge $\left(i,j\right)$, thus conflicting to the goal of fairness attacks. So, it should not be selected by FATE. Similarly, if we have negative $\Delta_{\mathbf{A}}b$ and $\left(\mathbf{I}-2\mathbf{A}\right)\left[i,j\right]<0$, removing edge $\left(i,j\right)$ would help increase the edge. However, $\left(\mathbf{I}-2\mathbf{A}\right)\left[i,j\right]>0$ suggests adding the edge. It is also conflicting to the goal of fairness attacks, and FATE would not select it.
>
> **Q3. Justification of applying kernel density estimation on non-IID graph data.**
>
> Thank you for your comments. To date, it is an open problem whether the learned node representations follow IID assumption on the low-dimensional manifold or not. Empirically from the experimental results, using KDE-based bias approximation effectively helps maximize the bias for fairness attacks. Meanwhile, relaxing the IID assumption is a common strategy in computing the distributional discrepancy of node representations. For example, MMD is a widely used distributional discrepancy measures, whose accurate approximation also requires IID assumption[1]. And recent studies[2,3] show that we can also adapt it on non-IID data which shows promising empirical performance.
>
> **Q4. Modifying DICE to inject/delete edges based on sensitive attributes.**
>
> Thank you for suggesting this baseline method. We have implemented it and evaluated its performance in attacking statistical parity. The additional results are provided in Table 2 of the one-page PDF. From the table, we observe that DICE-S could fail the fairness attacks in many cases. In other cases, DICE-S is less effective than FATE.
>
> **Q5. Discussions about related works[4,5]**
> Thank you for bringing up these two wonderful works. For [4], it samples the perturbation matrix for discretized attack and utilizes projected gradient descent-based method, which can be seen as an alternative to our preference matrix-based perturbation matrix generation. We would be happy to add a remark in our revised version to highlight such alternative strategy.
> For [5], in the context of fairness attacks, it would be interesting to explore whether we can generate a bias-augmented graph, which may empower us to attack multiple group fairness definitions simultaneously. We would be happy to discuss this potential future directions in our revised version. We would like to also point out that we assume the attacker has its own targeted fairness definition to attack. For example, the attacker aims to prevent certain demographic groups from getting financial support. And FATE is consistent with the setting. The setting suggested by [5] can be seen as more challenging than the current one.
>
> **Q6. Inconsistent budget constraints in line 98 and line 121.**
>
> Thank you for your comments. We believe that these two constraints are the same. In line 98, we use e.g. (for example) to provide an example, whereas in line 121, we use i.e. (in another word) to explain the distance constraints for edges/features.
>
> **Q7. Explanation for $\|\mathbf{A}\|_{1,1}$.**
>
> Thank you for your comments. For the induced matrix norms, $\left\|\mathbf{A}\right\|_{1,1}$ would refer to the maximum absolute column sum of the matrix difference. However, we use the entry-wise matrix norms. It treat the elements of the matrix as one big vector (see [here](https://en.wikipedia.org/wiki/Matrix_norm#%22Entry-wise%22_matrix_norms) and [here](https://www.cs.ubc.ca/~schmidtm/Courses/Notes/norms.pdf)). Following the definition of entry-wise matrix norms, our notation means the absolute sum of each entry in $\mathbf{A}$.
>
> **Q8.Typos about $\|\mathbf{A},\mathbf{\widetilde A}\|_{1,1}$ and Table 2.**
>
> For $\|\mathbf{A},\mathbf{\widetilde A}\|_{1,1}$, we have corrected it to $\|\mathbf{A}-\mathbf{\widetilde A}\|_{1,1}$. We have also corrected the metric in Table 2.
>
> **Q9.Inconsistent representation of bias function in line 94 and line 99.**
>
> Thank you for pointing it out. We have corrected it into $b\left(\mathbf{Y},\Theta^*,\mathbf{F}\right)$.
>
> ```
> [1] Badr-Eddine Chérief-Abdellatif, and Pierre Alquier. MMD-Bayes: Robust Bayesian estimation via maximum mean discrepancy. AABI2020.
> [2] Qi Zhu, Natalia Ponomareva, Jiawei Han, and Bryan Perozzi. Shift-robust gnns: Overcoming the limitations of localized graph training data. NeurIPS2021.
> [3] Qi Zhu, Yizhu Jiao, Natalia Ponomareva, Jiawei Han, and Bryan Perozzi. "Explaining and Adapting Graph Conditional Shift." arXiv:2306.03256.
> [4] Lu Lin, Ethan Blaser, and Hongning Wang. Graph structural attack by perturbing spectral distance. KDD 2022.
> [5] Nan Wang, Lu Lin, Jundong Li, and Hongning Wang. Unbiased graph embedding with biased graph observations. WWW2022.
> ```

---

> > ### Comment · Reviewer_NH2N · 2023-08-17
> > **Thank Authors for the Responce**
> >
> > I appreciate the efforts from the authors to address some of my questions. I would like to know the authors' thoughts on two places remaining unclear to me.
> >
> > Followup on Q1, can the author provide rigorous proof to show by limiting the learning rate, it is ensured to meet the budget constraint (no matter whether it is a single-step or multi-step attack)? I would hope to hear more about this treatment, as it is not a common solution for constrained optimization problem.
> >
> > Followup on Q4, DICE-S is evaluated under group or individual fairness? How is DICE-S implemented (e.g., how many edges are added/removed)? Is there any explanations on why DICE-S is not good enough (as based on Figure 2, the behavior shares similar patterns)?
> >
> > I am open to increasing my score if the authors can clarify these two questions.

---

> > > ### Author Response · Authors · 2023-08-17
> > > **Response to your follow-up questions**
> > >
> > > We appreciate your time and efforts in evaluating our work!
> > >
> > > **1. Proof for perturbation budget in continuous attack.**
> > >
> > > For a single-step attack, let $\mathbf{\widetilde A} = \mathbf{A} - \eta \nabla_{\mathbf A} b$. Then we have
> > > $$
> > > \|\|\mathbf{\widetilde A}-\mathbf{A}\|\|\_{1,1}
> > > = \eta \|\| \nabla_{\mathbf A} b \|\|\_{1,1}
> > > \leq \frac{B}{\|\| \nabla_{\mathbf A} b \|\|\_{1,1}} \|\| \nabla_{\mathbf A} b \|\|\_{1,1}
> > > = B
> > > $$
> > >
> > > The inequality in the middle holds because $\eta \leq \frac{B}{\|\| \nabla_{\mathbf A} b \|\|_{1,1}}$. Similarly, for multi-step case, following the same procedures, we can prove that the perturbation budget in the $i$-th step satisfies $\|\|\mathbf{\widetilde A}\_i -\mathbf{\widetilde A}\_{i-1}\|\|\_{1,1} \leq \delta\_i$ where $\mathbf{\widetilde A}\_0 = \mathbf{A}$, $\mathbf{\widetilde A}\_i$ is the poisoned adjacency matrix in the $i$-th step, and $\delta\_i$ is the budget in the $i$-th step. Then suppose we poison the adjacency matrix by $k$ steps. It is easy to get that $\|\|\mathbf{\widetilde A}\_{k} -\mathbf{A}\|\|\_{1,1} \leq B$ since we will choose the perturbation budget that satisfies $\sum\_{i=1}^n \delta_i \leq B$.
> > >
> > > **2. Performance of DICE-S.**
> > >
> > > The results in the global response is attacking statistical parity with GCN being the victim model. We have also mentioned in the global response that we only report results in attacking statistical parity on GCN due to limited space.
> > >
> > > Regarding its implementation, we slightly modify the implementation of DICE in the [deeprobust package](https://github.com/DSE-MSU/DeepRobust/blob/master/deeprobust/graph/global_attack/dice.py). Specifically, by passing the demographic group membership as the ```labels``` parameter in ```def attack```, DICE-S will add edges with two nodes from the same demographic group (instead of adding edges with two nodes of different labels in original DICE) and remove edges with two nodes from different demographic groups (instead of removing edges with two nodes of the same label in original DICE). Then we will use the same experimental settings described in Appendix C.5 to run DICE-S. The perturbation rates of DICE-S are the same as all other compared methods (i.e., vary from 0.05 to 0.25 with a step size of 0.05).
> > >
> > > About its suboptimal performance, we believe you are referring to the behavior in Figure 1 for attacking group fairness. We found out, for Pokec-z and NBA, of all the perturbations made by DICE-S, around 50% of perturbed edges (regardless of adding or removing) connecting two nodes with the same label (regardless of majority/minority label), and the rest 50% are edges connecting two nodes with different labels. According to the FA-GNN paper [1], if we add edges that connect nodes with the same sensitive attribute + different labels (case ED in [1]) and edges that connect nodes with different sensitive attribute + same labels (case DE in [1]), then the fairness attacks are not targeted. We conjecture that by perturbing edges of cases EE + ED with the same probability and cases DE + DD with the same probability, the acceptance rate for the advantaged group might be decreased, and the acceptance rate for disadvantaged group might be increased at the same time. And the effect might be neutralized, causing $\Delta_{\text SP}$ to be decreased after perturbations. Differently, for FATE, in addition to add edges with the same sensitive attribute, it also significantly increase edges that connect nodes to the minority class guided by the gradient of bias function, which makes the fairness attacks more targeted than DICE-S.
> > >
> > > On a side note, we would like to mention that a rigorous analysis of which edge contributes the most to bias amplification would require exhaustively perturbing each edge and re-training GNN, which could be computationally prohibitive to be achieved.
> > >
> > > Please let us know if you have any further concerns, and we would be happy to discuss with you.
> > >
> > > ```
> > > [1] Hussain Hussain, Meng Cao, Sandipan Sikdar, Denis Helic, Elisabeth Lex, Markus Strohmaier, and Roman Kern. Adversarial Inter-Group Link Injection Degrades the Fairness of Graph Neural Networks. ICDM 2022.
> > > ```

---

> > > > ### Comment · Reviewer_NH2N · 2023-08-18
> > > > **Thank Authors for the Response**
> > > >
> > > > Thank the author for the informative response, especially on Q4, which makes a lot sense to me.
> > > >
> > > > For Q1, the proof demonstrates that the constrain can be indeed enforced by limiting the learning rate. Just one issue regarding multi-step update, if you rigorously write out the inequality by summing up the differences for each step, $\|A-A_t\| \leq \|A - A_1\|+\|A_1 - A_2\|+\cdots+\|A_{t-1}-A_t\|\leq B$, the first triangle inequality could be pretty loose which makes the solution $A_t$ sub-optimal (as the difference could be much less than B). Therefore, I suggested the author to remark [4] as an alternative solution to this constrained optimization problem.
> > > >
> > > > Meanwhile, this framework can be potentially used to evaluate possible fairness defense (unbiased graph learning [5]). I hope the author will specifically note this in the paper to inspire the community for further steps.
> > > >
> > > > Overall, my concerns are addressed and I raised my score.

---

> > > > > ### Author Response · Authors · 2023-08-18
> > > > > **Thank you for the kind suggestions**
> > > > >
> > > > > Dear Reviewer NH2N,
> > > > >
> > > > > Thank you so much for the kind suggestions and raising your score! Indeed, we acknowledge that the proof for multi-step might get looser by applying triangle inequality, and we agree with you that [4] is an alternative to the optimization problem and [5] could be a potential defense strategy. We will definitely add more discussions about [4] and [5] in the revised version.
> > > > >
> > > > > Thanks,
> > > > >
> > > > > Authors

---

> ### Author Response · Authors · 2023-08-17
> **Follow-up with Reviewer NH2N**
>
> Dear Reviewer NH2N,
>
> We would like to thank you for your thorough review and kind suggestions to improve our work. We have provided a point-point response to your suggestions. Particularly, we have explained the use of our math notations and your concern related to KDE, added discussions about the suggested related work and a new baseline method, and conducted a thorough proofreading to correct typos and/or grammatical errors. Could you please check our response and let us know if you have further questions?
>
> All the best,
>
> Authors

---

### Official Review · Reviewer_G7MB · 2023-07-06

**Soundness:** 2 fair
**Presentation:** 4 excellent
**Contribution:** 3 good
**Rating:** 5
**Confidence:** 3

**Summary:**

The authors propose an attacking framework called FATE. Existing research in algorithmic fairness aims to prevent bias amplification but neglects fairness attacks. This paper fills this gap by formulating the fairness attack problem as a bi-level optimization and introducing a meta-learning-based attack framework. The authors present two instantiated examples, demonstrating the expressive power of the framework in terms of statistical parity and individual fairness, and validate the model's capability of attacking fairness through experimental verification. The paper contributes by providing insights into adversarial robustness and the design of robust and fair graph learning models.

**Strengths:**

1. This paper is well-structured and easy to follow.
2. The originality of the article deserves emphasis as it formulates the fairness attack on graph data as a bi-level optimization problem. This novel approach contributes to understanding the resilience of graph learning models to adversarial attacks on fairness.
3. The experimental results show that the proposed method is capable of attacking fairness without decreasing too much on accuracy.

**Weaknesses:**

The motivation provided is somehow insufficient in persuading me. The authors state that “an institution that applies the graph learning models are often utility-maximizing” (line 82), which I totally agree with, but then concludes that “minimizing the task-specific loss function … for deceptive fairness attacks” (line 88). While I do agree that pursuing utility will lead to a preference for models with superior performance, and if the objective of the attack is to deceive victims into selecting an unfair learning model, then it does make sense to enhance the utility of the malicious model. But it’s not the case in this article, where the attack’s aim is to poison the graph data. Unlike models, we don't have much discretion when it comes to the data, and it is exceedingly challenging for me to envision a real-life scenario wherein an institution would discard the data due to unsatisfactory performance, as a more practical solution is data cleansing or a more capable model. Consequently, my concern is that, is it truly necessary to maintain the utility for “deception”, leaving aside that I’m not convinced that preserving the utility is necessary for successful deception. Please provide a more detailed explanation.

The experimental findings reveal the limitations of the attack's effectiveness. Although I agree that FATE is capable of attacking fairness and is relatively more stable, in 2 out of 3 datasets, FATE exhibits incompetence in reducing statistical parity compared to FA-GNN. Although the authors argue that the victim model achieved the best performance under the attack FATE, I’m skeptical about the cost-effectiveness of this trade-off.

**Questions:**

Please refer to the weakness.
Found typo: in line 62, A[j,:] should be A[,:j]?

**Limitations:**

Yes, the authors have addressed the limitations and potential negative societal impact of their work.

---

> ### Author Rebuttal · Authors · 2023-08-10
>
> We appreciate your constructive comments and valuable feedback for further improving our work. We summarize the main concerns and the point-to-point responses as follows.
>
> **Q1. Necessity of maintaining utility for deception.**
>
> Thank you for your comments about the necessity of mataining utility. When the fairness attacks are not deceptive, it is easy for a utility-maximizing institution to detect the abnormal performance of the graph learning model, which is trained on the poisoned data. Such abnormal performance may trigger the auditing toward model behaviors, which may further lead to the investigation on the data used to trained the model. With careful auditing, it is possible to identify fairness attacks as shown in our analysis of manipulated edges due to significantly more edges that have certain properties (e.g., connecting two nodes with a specific sensitive attribute value). Then to avoid the negative consequences caused by fairness attacks, the institution might take actions such as data cleansing which is also suggested by the reviewer. Note that we do not suggest the institution to discard the data if fairness attacks happen. How to deal with such attacks and the poisoned data should be solely the decision of the institution. However, if the fairness attacks are deceptive, it is much less possible to trigger the auditing by monitoring the model performance, which increases the chance that the fairness attacks could make long-lasting negative societal impacts. Finally, we would like to also point out that, in addition to mataining similar utility, we also want the poisoned graph is close to the original graph by setting a budget on $d\left(\mathcal{G}, \mathcal{\widetilde G}\right)$ for deception.
>
>
>
> **Q2. Incompetence in reducing statistical parity compared with FA-GNN.**
>
> Thank you for your comments. We acknowledge that FA-GNN could achieve more bias than FATE in several cases. However, please note that our goal is not only increasing the bias, but also making the fairness attacks deceptive by ensuring the similarity between poisoned graph and benign graph as well as achieving comparable performance in the downstream tasks. Our goal is aligned with our problem definition (Problem 1). By considering the utility-maximizing institution, we believe that FATE provides a promising solution to achieve deceptive fairness attacks, since it can consistently increase the bias (for successful fairness attacks) while achieving much more comparable utility compared with FA-GNN (for making the attacks deceivable). Notably, FA-GNN has much more decrease in micro F1 score (i.e., 0.9\% decrease for Pokec-n, 2.9\% decrease for Pokec-z, 3.3\% decrease for Bail) while FATE's utility is more comparable to the performance on benign graph (i.e., 0.8\% for Pokec-n, 0.1\% increase for Pokec-z, 1.0\% decrease for Bail). Consequently, a utility-maximizing decision maker might be deceived to choose the model trained by poisoned graph generated by FATE since it achieve higher micro F1 score.
>
>
>
> **Q3. Typos.**
>
> Thank you for pointing it out. We have corrected the typo. Meanwhile, we will conduct a thorough proofreading to further improve our paper's writing in the revised version.
>
>
>
> **Q4. Ethics review.**
>
> Thank you for raising your concerns about potential ethics review of our work. We would like to further point out that the ethics reviewers do not find any violation of the conference guidelines about ethics. The goal of our paper is to investigate the possibility of making the graph learning results more biased, in order to raise the awareness of such fairness attacks. Meanwhile, our experiments suggest that existing fair graph neural networks suffer from the fairness attacks, which further highlight the importance of designing robust and fair techniques to protect the civil rights of marginalized individuals. We acknowledge that, when used for commercial purpose, the developed technique might be harmful to individuals from certain demographic groups. To prevent this situation from happening, we will release our code under [CC-BY-NC-ND license](https://libguides.hartford.edu/c.php?g=887688\&p=6380447\#:~:text=CC%2DBY%2DNC%2DND%20or%20Creative%20Commons%20Attribution%20NonCommercial,or%20ever%20use%20it%20commercially.), which prohibits the use of our method for any commercial purposes, and explicitly highlight in our released code that any use of our developed techniques will consult with the authors for permission first.

---

> ### Author Response · Authors · 2023-08-17
> **Follow-up with Reviewer G7MB**
>
> Dear Reviewer G7MB,
>
> We would like to thank you for your thorough review and kind suggestions to improve our work. We have replied to your suggestions, where we justified why maintaining utility could lead to more deceptive fairness attacks, explained the comparison with FA-GNN, and discussed the negative societal impacts. Could you please check our response and let us know if you have further questions?
>
> All the best,
>
> Authors

---

### Official Review · Reviewer_Lw8H · 2023-07-06

**Soundness:** 2 fair
**Presentation:** 3 good
**Contribution:** 2 fair
**Rating:** 5
**Confidence:** 4

**Summary:**

The paper proposes a novel framework named Fate, which is capable of attacking any fairness definition on any graph learning model, as long as the corresponding bias function and the task-specific loss function are differentiable. Fate is equipped with the ability for either continuous or discretized poisoning attacks on the graph topology.studies. The paper provides insights into the adversarial robustness of fair graph learning and sheds light on designing robust and fair graph learning in future studies. The empirical evaluation on three benchmark datasets shows that Fate consistently succeeds in fairness attacks while being the most deceptive (achieving the highest micro Fl score) on semi-supervised node classification.

**Strengths:**

(1) This paper addresses an important problem in graph learning — fairness attacks. While previous work in the field focused on ensuring that bias in not perpetuated or amplified during the learning process, the proposed framework, FATE, allows for the study of adversarial attacks on fairness.

(2) FATE, a meta-learning based framework, is versatile and can be used to attack different fairness definitions and graph learning models.

(3)The experimental evaluation shows that the proposed framework can successfully attack statistical parity and individual fairness on real-world datasets with the ability for poisoning attacks on both graph topology and node features while maintaining the utility on the downstream task.

(4) This article is well-written and well-organized. It starts with an introduction that highlights the importance of fair graph learning and the need for resilience against adversarial attacks. Then it provides some background information and defines the problem of fairness attacks in graph learning. The paper then proposes the Fate framework as a solution to this problem, providing a detailed explanation of its design and mechanism. It also presents experimental results to evaluate the efficacy of Fate. Finally, the paper concludes with a summary of its contributions and future research directions. Overall, the writing logic is clear and easy to follow.

(5) The paper provides detailed information on how they implemented the proposed framework, including the optimization process, the selection of the bias function and the task-specific loss function, and the hyperparameter tuning process. Additionally, they provide a detailed description of their experimental setup, including the datasets used, the graph learning models, the evaluation metrics, and the implementation details of the Fate framework and other baselines. The authors also provide a thorough evaluation of the proposed framework through extensive experiments and analysis.


**Weaknesses:**

(1) One weakness of this paper could be the limited evaluation of the framework on only three benchmark datasets and one task (semi-supervised node classification).
 Further evaluations on various graph learning tasks and datasets could provide more insights into the effectiveness and generalizability of the proposed framework.

(2) The proposed Fate framework may not be effective in attacking other fairness definitions beyond statistical parity and individual fairness, and it may not work well on graph learning models with very large graphs.

(3) The paper assumes that the attacker has access to sensitive attributes of all nodes, which may not always be feasible in real-world scenarios.


**Questions:**

(1) Have you considered the potential ethical concerns of using the proposed framework to amplify the bias in graph learning models?

(2) Could you conduct more ablation studies to further prove the effectiveness of the proposed framework compared to other methods?


**Limitations:**

see comments above

---

> ### Author Rebuttal · Authors · 2023-08-10
>
> We appreciate your constructive comments and valuable feedback for further improving our work. We summarize the main concerns and the point-to-point responses as follows.
>
> **Q1. More evaluation on different graph learning tasks and datasets.**
>
> Thank you for your suggestions. We choose the task as node classification because this is the most common setting in literature about fair graph learning. For fair regression on graphs, it is still an open problem, and a suitable benchmarking dataset is also needed. For fair link prediction, adapting FATE could be nontrivial, because the structural attack might cause data leakage issue in link prediction (i.e., a test link might be am adversarial edge injected by FATE). A potential remedy might be applying FATE to manipulate the node features instead of graph topology, which is beyond the focus of our paper. We also provide FATE's performance on an additional dataset named NBA. The detailed results can be found in Table 1 of the one-page PDF. From the table, we can get similar observations in our submission: FATE is the only method that can consistently achieve effective and deceptive fairness attacks.
>
>
>
> **Q2. FATE may not be effective in attacking fairness beyond statistical parity and individual fairness.**
>
> Thank you for your comments. In our paper, we choose statistical parity and individual fairness because they are the two most widely studied fairness definitions. Per our response to reviewer BvDw, we have also discussed how to attack Rawls fairness with FATE. Basically, to attack Rawls fairness, we can set the bias function to be the loss of the best/worst group. It is worth noting that such attack is conceptually the same as adversarial attacks on the utility as shown in [2], but only focusing on a subgroup of nodes determined by the sensitive attribute rather than the validation set. We will add detailed discussion on how to attack Rawls fairness in the revised version.
>
>
>
> **Q3. FATE may not work well when the graph is large.**
>
> Thank you for your comments. Though FATE's one major limitation is the time complexity (as discussed in `D -- Limitations' in Section 3.2 of our submission), FATE does not have any constraint on the properties of input graph. And we believe FATE's performance would not be impacted by how large the graph is.
>
>
>
> **Q4. Attacker having access to sensitive attributes of all nodes may not be feasible in real-world scenarios.**
>
> Thank you for your comments. Please note that we follow the same assumption compared with existing works [1]. Regarding the real-world scenario, as pointed out from line 18 -- line 23, a malicious (or polarized) banker might have access to the demographic information of bank account holders and aims to degrade the accuracies of fraud detection for different demographic groups. This example is consistent with our setting because the malicious banker knows the sensitive information of all nodes (i.e., bank account holders) in the graph and tries to make the learning results more biased.
>
>
>
> **Q5. Potential ethical concerns.**
>
> Thank you for your concerns regarding the societal impacts of our paper. We will explicitly highlight the following statements about ethical impacts in the revised version: The goal of our paper is to investigate the possibility of making the graph learning results more biased, in order to raise the awareness of such fairness attacks. Meanwhile, our experiments suggest that existing fair graph neural networks suffer from the fairness attacks, which further highlight the importance of designing robust and fair techniques to protect the civil rights of marginalized individuals. We acknowledge that, when used for commercial purpose, the developed technique might be harmful to individuals from certain demographic groups. To prevent this situation from happening, we will release our code under [CC-BY-NC-ND license](https://libguides.hartford.edu/c.php?g=887688&p=6380447#:~:text=CC%2DBY%2DNC%2DND%20or%20Creative%20Commons%20Attribution%20NonCommercial,or%20ever%20use%20it%20commercially), which prohibits the use of our method for any commercial purposes, and explicitly highlight in our released code that any use of our developed techniques will consult with the authors for permission first.
>
>
>
> **Q6. More ablation study on FATE.**
>
> Thank you for your suggestions. In our formulation, the upper-level optimization and lower-level optimization are coupled together. The lower-level optimization problem learn the results with high utility using the input graph (the input graph will be a poisoned graph after the first step of attack), and the upper-level optimization problem uses such learning results to further poison the graph to maximize the bias. In that sense, the dependency between these two main components makes it unable to ablate different variants by removing certain part. We would highly appreciate if the reviewer could clarify the specific setting that needs to be tested, and we would be more than happy to provide additional results if possible.
>
> ```
> [1] Hussain Hussain, Meng Cao, Sandipan Sikdar, Denis Helic, Elisabeth Lex, Markus Strohmaier, and Roman Kern. Adversarial Inter-Group Link Injection Degrades the Fairness of Graph Neural Networks. ICDM 2022.
> [2] Daniel Zügner, Oliver Borchert, Amir Akbarnejad, and Stephan Günnemann. Adversarial attacks on graph neural networks via meta learning. ICLR 2019.
> ```

---

> ### Author Response · Authors · 2023-08-17
> **Follow-up with Reviewer Lw8H**
>
> Dear Reviewer Lw8H,
>
> We would like to thank you for your thorough review and kind suggestions to improve our work. We have replied to your suggestions by adding additional experimental results, explaining our choices of evaluation task (node classification, ablation studies), and discussing the limitation and ethical concerns of our work. Could you please check our response and let us know if you have further questions?
>
> All the best,
>
> Authors

---

### Official Review · Reviewer_AtfZ · 2023-07-06

**Soundness:** 3 good
**Presentation:** 2 fair
**Contribution:** 3 good
**Rating:** 7
**Confidence:** 2

**Summary:**

This paper presents a novel approach for introducing fairness attacks in graph learning, which is impressive. To address this issue, the article proposes an attack framework for graphs and conducts experiments on the classic GCN model. Compared to two baseline methods, DICE and FA-GNN, the proposed method is more effective in attacking graph neural networks.

The strengths of the paper include:

1. The research problem is novel and interesting. There is little prior work on attacking graph models, and this article is the first to define this type of problem.
2. The paper provides code for the proposed method, which makes it more reproducible.

The weaknesses of the paper includes:

1. Some of the content organization is not optimal, such as placing the descriptions of the baseline methods in the appendix. Including them in the main text would have made it more convenient for readers.
2. The article does not mention the limitations of the work. More discussions are required.

**Strengths:**

1. The research problem is novel and interesting. There is little prior work on attacking graph models, and this article is the first to define this type of problem.
2. The paper provides code for the proposed method, which makes it more reproducible.

**Weaknesses:**

1. Some of the content organization is not optimal, such as placing the descriptions of the baseline methods in the appendix. Including them in the main text would have made it more convenient for readers.
2. The article does not mention the limitations of the work. More discussions are required.

**Questions:**

How is the efficiency of the method when the graph is growing large, say, with millions of nodes?

**Limitations:**

The article does not mention the limitations of the work. More discussions are required.

---

> ### Author Rebuttal · Authors · 2023-08-10
>
> We appreciate your constructive comments and valuable feedback for further improving our work. We summarize the main concerns and the point-to-point responses as follows.
>
> **Q1. Sub-optimal content organization.**
>
> Thank you for your suggestions in content organization. In the revised version, we will add a brief description to discuss the general idea of each baseline method. However, please note that, due to limited space, we need to defer some detailed experimental settings in the appendix to comply with the submission guidelines.
>
>
>
> **Q2. Discussion about the limitation.**
>
> Thank you for your comments. We believe that the limitation of our proposed method has been included in the submission. Please refer to `D -- Limitations' in Section 3.2 of our submission for your reference. In our discussion, we have mentioned that one major limitation is the time complexity, which is shared by several related works as well. And we have provided a brief discussion about the potential remedy to the this limitation.
>
>
>
> **Q3. Efficiency of \textsc{Fate} when the graph grows to millions of nodes.**
>
> Thank you for your comments in testing the efficiency of FATE on million-scale graphs. As we pointed out in the discussion about FATE's limitations in our original submission and to your previous concern (Q2), calculating the gradient with respect to the adjacency matrix $\mathbf{A}$ would require $O\left(n^2\right)$ space complexity. Due to the limited computational resources, we cannot evaluate the performance of FATE on graphs with millions of nodes. Please note that addressing the time complexity remains an open problem shared by several existing works [1, 2, 3].
>
> ```
> [1] Daniel Zügner, Oliver Borchert, Amir Akbarnejad, and Stephan Günnemann. Adversarial attacks on graph neural networks via meta learning. ICLR 2019.
> [2] Wei Jin, Yao Ma, Xiaorui Liu, Xianfeng Tang, Suhang Wang, and Jiliang Tang. Graph structure learning for robust graph neural networks. KDD 2020.
> [3] Zhe Xu, Boxin Du, and Hanghang Tong. Graph sanitation with application to node classification. WWW 2022.
> ```

---

> ### Author Response · Authors · 2023-08-17
> **Follow-up with Reviewer AtfZ**
>
> Dear Reviewer AtfZ
>
> We would like to thank you for your thorough review and kind suggestions to improve our work. We have replied to your suggestions by re-organizing content and discussing the efficiency and limitation of our work. Could you please check our response and let us know if you have further questions?
>
> All the best,
>
> Authors

---

> > ### Comment · Reviewer_AtfZ · 2023-08-20
> > **Reply to author rebuttal**
> >
> > My concerns about the work are partially solved. Hope to see the supplemented contents in the paper. I have no further questions.

---

### Official Review · Reviewer_BvDw · 2023-07-24

**Soundness:** 3 good
**Presentation:** 3 good
**Contribution:** 3 good
**Rating:** 6
**Confidence:** 3

**Summary:**

This paper studies an interesting problem, attacking fairness on GNN. Specifically, the authors aim to amplify the unfairness while maintaining the performance of the downstream tasks. They propose a bi-level optimization scheme with a meta-gradient poisoning attack to achieve this goal. Experiments on both statistic fairness and individual fairness show the effectiveness of the proposed method. Overall, good problem and a solid framework.

**Strengths:**

1.	Fairness attack on graph learning is an interesting problem, this paper gives an intuitive problem definition, which can extend to other bi-level optimization goal attacking scenarios.
2.	Given the bi-level optimization goal, the authors design a meta-gradient graph poisoning attack and corresponding Low-level and High-level loss functions.
3.	Experiments on statistical fairness and individual fairness demonstrate the effectiveness of the proposed method.


**Weaknesses:**

1. It will be more interesting if the authors provide more experiments on group fairness and Rawls fairness. How to attack some specific groups, such as best/worst accuracy group fairness.

**Questions:**

It will be more interesting if the authors provide more experiments on group fairness and Rawls fairness. How to attack some specific groups, such as best/worst accuracy group fairness.

---

> ### Author Rebuttal · Authors · 2023-08-10
>
> We appreciate your constructive comments and valuable feedback for further improving our work. Please see our point-to-point responses as follows.
>
> **Q1. How to group/Rawls fairness by attacking a specific demographic group or best/worst accuracy group.**
>
> Thank you for your comments in adding more discussions about instantiation of other fairness definitions. Here, we briefly discuss how to attack a specific demographic group and best/worst accuracy group. We will add detailed discussions in the revised version.
>
> To attack a specific group, there can be two strategies: (1) decreasing the acceptance rate of the corresponding group and (2) increasing the gap between the group to be attacked and another demographic group. For (1), following our strategy of modeling acceptance rate as the CDF of Gaussian KDE, we can set the bias function to be maximize as the negative of acceptance rate (i.e., $b\left(\mathbf{Y},\Theta^*,\mathbf{F}\right)=-\operatorname{P}\left[\hat{y}=1\vert s=a\right]$) where $a$ is the sensitive attribute value denoting the demographic group to be attacked. For (2), suppose we want to attack the group with sensitive attribute value $s=0$. We can also attack this demographic group by setting the bias function to be $b\left(\mathbf{Y},\Theta^*,\mathbf{F}\right)=\operatorname{P}\left[\hat{y}=1\vert s=1\right]-\operatorname{P}\left[\hat{y}=1\vert s=0\right]$. In this way, we can increase the acceptance rate of demographic group ($s=1$) while minimizing the acceptance rate of the group ($s=0$).
>
> To attack Rawls fairness by attacking best/worst accuracy, the general idea could be setting the bias function to be the loss of the best/worst group. It is worth noting that such attack is conceptually the same as adversarial attacks on the utility as shown in [1], but only focusing on a subgroup of nodes determined by the sensitive attribute rather than the validation set.
>
> ```
> [1] Daniel Zügner, Oliver Borchert, Amir Akbarnejad, and Stephan Günnemann. Adversarial attacks on graph neural networks via meta learning. ICLR 2019.
> ```

---

> ### Author Response · Authors · 2023-08-17
> **Follow-up with Reviewer BvDw**
>
> Dear Reviewer BvDw,
>
> We would like to thank you for your thorough review and kind suggestions to improve our work. We have replied to your concerns by discussing about how to attack a specific group for group fairness and how to attack best/worst accuracy group for Rawls fairness. Could you please check our response and see if you have further questions/suggestions?
>
> All the best,
>
> Authors

---

### Author Rebuttal · Authors · 2023-08-10

To all reviewers,

We sincerely thank your time and efforts in evaluating our work. In our rebuttal, we have provided point-to-point responses to your concerns. Meanwhile, in this thread, we provide this one-page PDF that includes additional experimental results on a new dataset named NBA and an new baseline method.

If you have any questions regarding our responses, we would be more than happy to discuss with you and further clarify them.

Thank you,
Authors

---

> ### Comment · Reviewer_NH2N · 2023-08-17
> **PDF not found**
>
> Thank the authors for the detailed response. I did not find the one-page PDF under this thread. Let me know if I missed anything.

---

> > ### Author Response · Authors · 2023-08-17
> > **Sorry for the missing PDF**
> >
> > Dear Reviewer NH2N,
> >
> > We are sorry about the missing PDF. This might be due to poor internet connection at the time of response. The experimental results for DICE-S (i.e., DICE by removing/adding edges based on sensitive attribute) are as follows, which will be added in our revised version. In each cell, the results are reported when the perturbation are from 0.00 to 0.25 with a step size of 0.05 in an ascending order, i.e., the leftmost value indicates the result when the perturbation rate is 0.00 and the rightmost value indicates the result when the perturbation rate is 0.25.
> >
> > |Dataset|Micro-F1|Delta_SP|
> > |-|-|-|
> > |Pokec-n|68.3&rarr;67.6&rarr;67.9&rarr;67.4&rarr;66.1&rarr;65.9|6.8&rarr;7.1&rarr;7.2&rarr;7.9&rarr;7.1&rarr;6.5|
> > |Pokec-z|68.7&rarr;68.8&rarr;67.7&rarr;67.8&rarr;67.0&rarr;67.4|6.3&rarr;5.7&rarr;6.5&rarr;4.8&rarr;5.9&rarr;5.8|
> > |Bail|92.5&rarr;92.3&rarr;92.2&rarr;90.0&rarr;91.8&rarr;91.6|8.2&rarr;8.4&rarr;8.5&rarr;8.4&rarr;9.1&rarr;9.3|
> > |NBA|70.0&rarr;71.8&rarr;71.3&rarr;72.8&rarr;75.1&rarr;71.3|15.5&rarr;11.0&rarr;5.1&rarr;14.9&rarr;12.2&rarr;10.7|
> >
> > From the experimental results, we can see that DICE-S is not outperforming the proposed FATE framework. It also fails to achieve fairness attacks for Pokec-n (when perturbation rate is 0.25), Pokec-z (except for the case when perturbation rate is 0.1), and NBA (in all cases).
> >
> > Please let us know if you have further questions about the results.
> >
> > All the best,
> >
> > Authors

---

> ### Author Response · Authors · 2023-08-17
> **Sorry about the missing PDF**
>
> Dear reviewers,
>
> We are sorry about the missing PDF. This might be due to poor internet connection at the time of response. Please see our additional results in the PDF as follows, including the experimental results of attacking statistical parity on GCN (1) on NBA dataset (as asked by Reviewer Lw8H) and (2) for DICE-S (i.e., DICE by removing/adding edges based on sensitive attribute as asked by Reviewer NH2N), both of which will be added in our revised version.
>
> In each cell, the results are reported when the perturbation are from 0.00 to 0.25 with a step size of 0.05 in an ascending order, i.e., the leftmost value indicates the result when the perturbation rate is 0.00 and the rightmost value indicates the result when the perturbation rate is 0.25.
>
> ```Experimenta results for NBA```
> |Method|Micro-F1|Delta_SP|
> |-|-|-|
> |Random|70.0&rarr;67.1&rarr;68.5&rarr;70.0&rarr;69.5&rarr;68.7|15.5&rarr;10.2&rarr;5.2&rarr;11.0&rarr;4.8&rarr;9.2|
> |DICE|70.0&rarr;69.5&rarr;70.0&rarr;68.7&rarr;75.1&rarr;84.1|15.5&rarr;8.4&rarr;7.8&rarr;8.3&rarr;10.6&rarr;21.4|
> |FA-GNN|70.0&rarr;71.3&rarr;70.8&rarr;72.1&rarr;69.2&rarr;69.0|15.5&rarr;24.6&rarr;28.7&rarr;38.5&rarr;11.6&rarr;12.6|
> |FATE-flip|70.0&rarr;70.8&rarr;66.2&rarr;68.2&rarr;66.9&rarr;64.9|15.5&rarr;30.6&rarr;39.1&rarr;30.6&rarr;36.7&rarr;45.3|
> |FATE-add|70.0&rarr;69.2&rarr;71.5&rarr;66.9&rarr;66.4&rarr;64.6|15.5&rarr;31.7&rarr;27.5&rarr;33.7&rarr;40.0&rarr;43.6|+
>
> From the above results, we have the following observations: (1) Both FATE-flip and FATE-add can successfully attack fairness regardless of the choice of victim model; (2) FATE-flip achieves more deceptive fairness attacks than FA-GNN when perturbation rate is 0.20 by providing higher utility while amplifying bias at the same time, and FATE-add achieve more deceptive fairness attacks than FA-GNN when  the perturbation rate is 0.10; (3) When the perturbation rate is equal to or higher than 0.2, FA-GNN is unable to achieve fairness attacks; (4) And DICE fails to acheive fairness attacks, except for its surprisingly well performance when the perturbation rate is 0.25. As a short summary, our main conclusion remains the same: FATE is the only method that can consistently achieve effective and deceptive fairness attacks.
>
> ```Experimental results for DICE-S```
> |Dataset|Micro-F1|Delta_SP|
> |-|-|-|
> |Pokec-n|68.3&rarr;67.6&rarr;67.9&rarr;67.4&rarr;66.1&rarr;65.9|6.8&rarr;7.1&rarr;7.2&rarr;7.9&rarr;7.1&rarr;6.5|
> |Pokec-z|68.7&rarr;68.8&rarr;67.7&rarr;67.8&rarr;67.0&rarr;67.4|6.3&rarr;5.7&rarr;6.5&rarr;4.8&rarr;5.9&rarr;5.8|
> |Bail|92.5&rarr;92.3&rarr;92.2&rarr;90.0&rarr;91.8&rarr;91.6|8.2&rarr;8.4&rarr;8.5&rarr;8.4&rarr;9.1&rarr;9.3|
> |NBA|70.0&rarr;71.8&rarr;71.3&rarr;72.8&rarr;75.1&rarr;71.3|15.5&rarr;11.0&rarr;5.1&rarr;14.9&rarr;12.2&rarr;10.7|
>
> From the table, we can see that DICE-S is not outperforming the proposed FATE framework. It also fails to achieve fairness attacks for Pokec-n (when perturbation rate is 0.25), Pokec-z (except for the case when perturbation rate is 0.1), and NBA (in all cases).
>
> ```
> NOTE: To comply with NeurIPS submission rule of adding only one-page PDF and to be fair with all other submissions, we only report results in attacking statistical parity on GCN due to limited space, and we only copy our experimental results in the PDF to this thread. In our revised version, we will add additional experiment results in all experimental settings (including attacking individual fairness and setting the victim model as FairGNN and InFoRM-GNN).
> ```

---

### Comment · Area_Chair_SpQz · 2023-08-18
**Wait for Reviewers' Feedback**

Dear Reviewer BvDw, AtfZ, Lw8H, and G7MB,

The authors and I are eager to know whether the author responses successfully address your concern. The Author-Reviewer discussion ends on **August 21**. You are strongly encouraged to directly reply to the authors.

Thank you for your hard work.

PS: (1) This is a public thread. (2) I am expecting to know your thoughts as well. If you want to individually reply to me, please use the thread of internal discussion.

Kind Regards,

AC

---

### Decision · Program_Chairs · 2023-09-21

**Decision:**

Reject

**Comment:**

I have read all the materials of this paper including the manuscript, appendix, comments, and response. Based on collected information from all reviewers and my personal judgment, I can make the recommendation on this paper, rejection. No objection from reviewers who participated in the internal discussion was raised against the rejection recommendation.

**Research Problem**

This paper addresses the fairness attach on graphs.

**Motivation**

Line 29-33 illustrates two drawbacks of existing methods in the fairness attack category. (1) Adversarial data injection methods are often designed for tabular data. This is not true. The authors provide a paragraph on adversarial attacks on graphs in the related work. Therefore, the authors need to illustrate the drawbacks of literation on adversarial attacks on graphs, especially what the research challenges are to extend the current methods for tackling fairness attack. (2) The current method only attacks group fairness. The authors believe it is crucial to study the attack of different notations of fairness for a variety of graph learning models. More comprehensiveness does not mean better. If a paper considers the group fairness, which is its research question, it is improper to criticize it cannot handle individual fairness. Just like this paper that can only tackle the differentiable bias functions, but I do not regard this point as a drawback. If some simple extensions can achieve comprehensiveness, it does not count for novelty or contributions. In this paper, the authors also consider the individual fairness. Since there is a huge body of literature on individual fairness on graphs, it also needs to discuss the difficulty of extending these works in the attack scenario.

I did notice the third motivation in Line 80, which considers attacking fairness and preserve the algorithmic utility. This is good.

**Philosophy**

For the above first point, the challenge is not specific, and I did not find any philosophy to tackle this. For the second point, the authors employed the bi-level optimization for generalization on fairness notations and graph models, where the third point is also embedded as a constraint.

**Related Work**

Since the authors also address the individual fairness, it would be to provide a paragraph on that.

**Technique**

Respectfully, I did not see too much innovation in the technical part. Moreover, the lower-level optimization is problematic, which minimizes the loss function towards utility. As the authors mentioned, performance degradation in utility makes the attack deceptive. However, utility increase will also raise the deceptiveness. Is it better that preserving the original utility and degrading the fairness?

**Experiments**

1. Since there are many graph attack methods, why the authors only compare with DICE?
2. It is suggested to provide the significant analysis.
3. In Table 1, FA-GNN achieves the best performance in $\delta_sp$, but not bold.
4. I noticed some utility gain of FATE in Table 1, which makes the model deceptive (See my point above).